# A novel MCGDM technique based on correlation coefficients under probabilistic hesitant fuzzy environment and its application in clinical comprehensive evaluation of orphan drugs

**Yubo Hu** \*, **Zhiqiang Pang**

School of Statistics, Lanzhou University of Finance and Economics, Lanzhou, Gansu, China

\* huyubo202209@163.com

## Abstract

Probabilistic hesitant fuzzy sets (PHFSs) are superior to hesitant fuzzy sets (HFSs) in avoiding the problem of preference information loss among decision makers (DMs). Owing to this benefit, PHFSs have been extensively investigated. In probabilistic hesitant fuzzy environments, the correlation coefficients have become a focal point of research. As research progresses, we discovered that there are still a few unresolved issues concerning the correlation coefficients of PHFSs. To overcome the limitations of existing correlation coefficients for PHFSs, we propose new correlation coefficients in this study. In addition, we present a multi-criteria group decision-making (MCGDM) method under unknown weights based on the newly proposed correlation coefficients. In addition, considering the limitations of DMs' propensity to use language variables for expression in the evaluation process, we propose a method for transforming the evaluation information of the DMs' linguistic variables into probabilistic hesitant fuzzy information in the newly proposed MCGDM method. To demonstrate the applicability of the proposed correlation coefficients and MCGDM method, we applied them to a comprehensive clinical evaluation of orphan drugs. Finally, the reliability, feasibility and efficacy of the newly proposed correlation coefficients and MCGDM method were validated.

## 1. Introduction

In recent years, rare diseases have become a significant public health concern worldwide. Orphan medications are used to diagnose, prevent, or treat rare disorders. In general, health technology assessment (HTA) plays an essential role in a country's drug procurement, drug reimbursement policy, and drug price decisions as a key technological method for comprehensive clinical evaluation of pharmaceuticals. However, owing to a lack of appropriate clinical trial data, the therapeutic value and economic evaluation of orphan pharmaceuticals are difficult to measure using typical drug standards, making it extremely difficult to utilize traditional

**Funding:** The author(s) received no specific funding for this work.

**Competing interests:** The authors have declared that no competing interests exist.

HTA to completely evaluate orphan drugs. As a result, conducting reasonable clinical comprehensive evaluations of orphan medications is a difficult problem faced by all countries, making it critical to investigate effective clinical evaluation methodologies for orphan drugs. Many researchers have increasingly included the multi-criteria decision-making (MCDM) method for the comprehensive clinical evaluation of orphan medications in recent years [1–3]. Unlike traditional HTA, which focuses on the clinical comprehensive evaluation of pharmaceuticals using a single criterion of cost-benefit analysis, MCDM can be utilized for the thorough clinical evaluation of drugs using several dimensions. In comparison to HTA, MCDM is more suitable for the comprehensive clinical evaluation of orphan medications. However, MCDM approaches employed in the comprehensive clinical assessment of orphan medicines have some limitations. In fact, owing to the low prevalence of uncommon diseases, the number of patients is minimal, and clinical trial data are lacking. Consequently, the evaluation of orphan medications is primarily based on the subjective opinions of experts. Second, because of the ambiguity and uncertainty of the evaluated objects, the limitations brought about by experts' different knowledge, experience, and cognition, and the hesitation shown by experts when evaluating multiple evaluation values, existing MCDM methods for the comprehensive clinical evaluation of orphan drugs do not consider uncertainty, ambiguity, and hesitation in the expert decision-making process. Expert evaluation information cannot be accurately expressed using simple expert scoring and subjective weighting. The comprehensive clinical evaluation of orphan drugs is a typical fuzzy MCDM problem. The fuzzy theory-based MCDM method helps deal with uncertainty, ambiguity, and hesitancy in decision making. As a result, choosing an assessment information expression form that conforms to the expert thinking process and studying effective fuzzy MCDM approaches will increase the accuracy of the comprehensive clinical evaluation of orphan medications. However, some research gaps remain in the study of fuzzy MCDM difficulties in the comprehensive clinical assessment of orphan drugs: 1. Although probabilistic hesitant fuzzy sets (PHFSs) have been used and validated by scholars in many application scenarios, few scholars have introduced PHFS into the clinical comprehensive evaluation of orphan drugs; 2. In fuzzy sets, the quality of the information measurement of fuzzy sets determines the effectiveness of the fuzzy MCDM methods. There are currently certain study gaps in the research on correlation coefficients in PHFSs. 3. Medical experts typically view the comprehensive clinical evaluation of orphan pharmaceuticals as a group decision-making process. Second, medical professionals evaluate diverse medications using language factors. However, transforming the linguistic variable decision information provided by each expert into a form that can reflect the fuzziness, hesitancy, and importance of evaluation values is a worthwhile research question; 4. Furthermore, despite the fact that MCDM methods in probabilistic hesitant fuzzy (PHF) environments have been widely studied, there are still some research gaps in the research on correlation coefficients, implying the need for further improvement of existing MCDM methods in PHF environments.

Based on the above research motivation, this study first conducted a detailed study on the correlation coefficient for PHFSs, proposed some new correlation coefficients, and considered that decision-makers(DMs) are accustomed to using linguistic variables when evaluating various criteria. We then proposed a method to convert linguistic variables into probabilistic hesitant fuzzy information. Based on the above research, we further proposed a correlation coefficient-based multi-criteria group decision-making (MCGDM) method under a PHF environment with unknown weights. Finally, we demonstrated our proposed method through a case study of orphan drug evaluation.

## 2. Literature review

At present, the MCDM methods most commonly utilized in the clinical comprehensive assessment of orphan medications are expert scoring [1, 2, 4–6], simple additive weighing [7–9], and analytic hierarchy processes [10, 11]. These methods mostly rely on expert subjective judgment to provide decision results by directly scoring relevant features or assigning weights to the criteria. In fact, each expert faces uncertainty when scoring criteria owing to the ambiguity and uncertainty of evaluating things themselves, as well as the limitations brought about by experts' differing knowledge, experience, and cognition, and the hesitation shown when experts evaluate multiple evaluation values. Existing orphan drug evaluation approaches do not account for uncertainty, ambiguity, or hesitation in the expert decision-making process. Expert evaluation information cannot be accurately expressed by relying solely on subjective evaluation. The comprehensive clinical evaluation of orphan drugs is a typical fuzzy MCDM problem. Consequently, the key to conducting a comprehensive clinical assessment of orphan drugs is to use an effective expression form that can express the ambiguity, uncertainty, and hesitation of experts' evaluation information.

Therefore, in view of the above situation, Zadeh [12] proposed fuzzy sets and their extended forms, such as L-type fuzzy sets [13], 2-type fuzzy sets [14], fuzzy interval sets [15], and intuitionistic fuzzy sets [16]. Although the aforementioned extended forms of fuzzy sets have helped DMs deal with the majority of decision application scenarios to a certain extent, researchers have discovered that DMs hesitate between multiple degrees of membership. Therefore, Torra [17] proposed hesitant fuzzy sets(HFSs), and extended forms of HFSs, such as dual hesitant fuzzy sets [18] and interval-valued hesitant fuzzy sets [19], have been extensively applied to group decision-making processes. However, similar to the fuzzy set and its other extension forms, HFSs also have some shortcomings, which are primarily manifested by the fact that HFSs ignore DMs' preferences information, preventing them from expressing their preferences in their entirety. Consequently, Xu and Zhou [20] incorporated probability information into HFSs and proposed probabilistic hesitant fuzzy sets (PHFSs) to circumvent the issue of DMs' preference information loss of DMs in HFSs. Subsequently, corresponding extended forms have been proposed [21–23]. For instance, Zhang et al. [21] proposed probabilistic interval-valued hesitant fuzzy sets and Liu [22] proposed probabilistic linguistic term sets. Hao et al. [23] introduced the concept of probabilistic dual hesitant fuzzy sets. In recent years, an increasing number of researchers have focused on PHFSs, including their fusion operator [24–26], preference relationships [27, 28], measures based on PHFSs [29–32], and decision methods based on PHFSs [32–35], etc. Although PHFSs have been extensively investigated, certain unresolved issues remain.

The correlation coefficient has been extensively utilized in numerous applications, including data analysis and classification, pattern recognition, and decision-making [36–39], etc., as a tool for measuring the degree of linear correlation between random variables in statistics. As the decision-making environment becomes increasingly uncertain, the concept of correlation coefficient has been applied to fuzzy environments [40–52]. Gerstenkorn and Manko [40] were the first to introduce a correlation coefficient to intuitionistic fuzzy sets. Based on intuitionistic fuzzy sets, Bustince and Burillo [41] proposed a correlation coefficient of interval-valued intuitionistic fuzzy sets. Hong and Hwang [42] investigated the correlation coefficients of intuitionistic fuzzy sets in a probability space. Ye [44] proposed a weighted correlation coefficient based on entropy weight in an intuitionistic fuzzy environment and applied it to MCDM. Chen et al. [45] presented a number of correlation coefficients for a hesitant fuzzy environment and employed them in cluster analysis. Ye [46] proposed a correlation coefficient for dual hesitant fuzzy sets based on HFSs and intuitionistic fuzzy sets. Liao et al. [47] pointed

out the inadequacy of traditional correlation coefficients in fuzzy sets, intuitionistic fuzzy sets, HFSs, etc., and proposed a new correlation coefficient, which was extended to the hesitant fuzzy linguistic term set. Singh [48] proposed the correlation coefficient of picture fuzzy sets, considering positive, neutral, negative, and rejected membership. Garg [49] pointed out the weakness of the existing correlation coefficient between intuitionistic fuzzy sets, and proposed a new correlation coefficient and weighted correlation coefficient formula to measure the relationship between two intuitionistic fuzzy sets. Considering that T-sphere fuzzy sets are an extension of fuzzy sets, intuitionistic fuzzy sets, and picture fuzzy sets, Ullah et al. [51] noted that the correlation coefficient of intuitionistic fuzzy sets and picture fuzzy sets is not applicable in some cases, and proposed the correlation coefficient of T-sphere fuzzy sets, which is used in clustering and multi-criteria decision making. Unlike HFSs, PHFSs include probabilistic information to compensate for the absence of information loss in the DMs. Therefore, numerous researchers have focused on the correlation coefficients of PHFSs [53–55]. Wang and Li [53] proposed the correlation coefficient of probabilistic hesitant fuzzy elements (PHFEs) by employing the concepts of mean value and covariance, without considering the duration of the PHFEs. They then proposed a weighted correlation coefficient of PHFEs and a weighted correlation-based MCDM method. Song et al. [54] also utilized the concept of mean value and variance to propose two types of correlation coefficients to measure the relationship between PHFSs without considering the length of PHFEs, and used the correlation coefficients in cluster analysis. From the aforementioned studies on correlation coefficients, it is evident that researchers construct correlation coefficients by referring to the correlation coefficient formulas in statistics, and then introducing the mean value and variance into PHFSs or PHFEs without considering the number of elements between PHFEs. However, we found that if the mean of each corresponding PHFE between multiple different PHFSs is the same, the conclusion that the correlation coefficient between multiple PHFSs is the same will be reached, which is inconsistent with the definition of the correlation coefficient of PHFS. Liu and Guan [55] proposed a hybrid correlation coefficient for a PHFS under these circumstances. Although this new correlation coefficient eliminates the aforementioned flaws, we found that calculating it is complicated and time-consuming. Second, the correlation coefficient is dependent on the weight setting of the mean, variance, and length rate correlation coefficients, which makes the calculation of the correlation coefficient somewhat subjective. When the correlation coefficient is applied to a decision, the resulting decision may also be subjective. Therefore, it is necessary to enhance and investigate the correlation coefficients of the PHFSs so that they can be applied to a wider variety of situations. Dumitrescu [56] introduced the concept of information energy to fuzzy sets. Gerstenkorn and Manko [40] subsequently extended the information energy to intuitionistic fuzzy sets and introduced the correlation coefficient of the intuitionistic fuzzy sets. Bustince and Burillo [41] extended the information energy to interval-valued intuitionistic fuzzy sets and proposed correlation coefficients. In addition, Chen et al. [45] cited the works of the aforementioned researchers, introduced information energy into HFSs, and proposed a correlation coefficient for HFSs. Based on the aforementioned research, this study incorporates information energy into PHFSs and proposes several new correlation coefficients for PHFSs, considering the length of the PHFEs. In addition, we consider the weights and propose several weighted correlation coefficients to overcome the deficiencies of the existing correlation coefficients of PHFSs.

Since simple additive weighting (SAW) was established by MacCrimmon [57] in the 1960s, MCDM approaches have been intensively explored as one of the decision-making strategies. These methods were initially developed under deterministic conditions, where the criterion values are expressed as real numbers. These include pairwise comparison methods, such as AHP [58], ANP [59], BWM [60], DEMATEL [61], and RANCOM [62]; outranking

methods, such as ELECTRE [63] and PROMETHEE [64]; reference point-based methods, such as TOPSIS [65], VIKOR [66], and EDAS [67]; utility-based methods, such as MOORA [68], MULTIMOORA [69], COPRAS [70], and SMART [71], as well as COMET [72], ESP-COMET [73], SPOTIS [74], SIMUS [75], and other methods proposed in recent years to overcome the phenomenon of rank reversal. Each of these methods has benefits and drawbacks, and there is no single best or worst method, which approach is used is determined by the DM's preferences and the necessities of the decision scene. As the decision-making environment faced by DMs becomes increasingly complex, it is no longer possible to meet the needs of decision-making by relying only on the decision in a deterministic setting. With the proposal of fuzzy sets, many scholars have gradually combined fuzzy set theory and MCDM methods to propose corresponding fuzzy MCDM methods in hesitant fuzzy environments [76], intuitive fuzzy environments [77], single-valued neutrosophic environments [78], and spherical fuzzy environments [79]. In recent years, with the proposal of PHFSs, compared to other fuzzy sets, they have excellent advantages in preserving DMs' preferences information. Many scholars have begun to study MCDM methods in probabilistic hesitant fuzzy environments [30, 53, 80–82]. Most fuzzy MCDM approaches rely on information measurements, such as distance, similarity, and correlation coefficients. Because of its simple computation steps and low processing complexity when compared to other reference point methods, the MCDM approach based on the correlation coefficient, as an extension of the reference point method, has been explored and developed in many fuzzy contexts [83–88]. However, the correlation coefficient determines the quality of the final decision-making method. Given the shortcomings of the existing correlation coefficient research in PHFSs, it is necessary to investigate the fuzzy MCDM method based on correlation coefficients in probabilistic hesitant fuzzy environments.

The primary contributions of this study: 1. Considering the limitations of the existing correlation coefficients of PHFSs, we incorporated information energy into PHFSs and proposed a series of new correlation coefficients and weighted correlation coefficients of PHFSs; 2. We proposed a PHF-MCGDM method based on the newly proposed correlation coefficient under unknown weights; 3. In this newly proposed PHF-MCGDM method, considering the limitations of DMs' habit of using language variables for expression in the evaluation process and inspired by Chen and Xu [89], we proposed a method for transforming the evaluation information of language variables into PHF values. Based on the above research, we applied the newly proposed MCGDM method for the comprehensive clinical evaluation of orphan drugs.

The remaining sections of this article are structured as follows:

In the third section, we review the concepts of HFSs and PHFSs, as well as their respective correlation coefficients, before discussing the deficiencies in the extant correlation coefficients for PHFSs. The fourth section offers and shows a range of correlation coefficient formulations and weighted versions and demonstrates their properties. In the fifth section, we present a method for converting linguistic variable assessment information into PHF information. We obtained the PHF group decision matrix as well as the criteria weights using this method. Finally, we apply the previously described correlation coefficient to the MCGDM and propose a novel MCGDM method based on the PHFS correlation coefficient with undetermined weights. In the sixth section, we apply the newly suggested MCGDM approach to the comprehensive clinical evaluation of orphan drugs to illustrate the applicability of our proposed method. The reliability, practicality, and validity of the proposed correlation coefficients and corresponding MCGDM method are examined in the seventh part. In the eighth section, we present a summary of this study and look ahead for future research.

## 3. Preliminaries

In this section, we review the concepts related to HFSs, PHFSs, and their corresponding correlation coefficients.

### 3.1. The related concept of HFS

To solve the problem of group decision making and the situation in which DMs are hesitant to face multiple membership degrees, Torra [17] introduced the HFS concept.

**Definition 1**. [17] Let X be a fixed set; then, the HFS is a function that maps every element of X to a subset of [0,1], and the mathematical expression of the HFS is as follows:

$$A = \{\langle x, h_A(x)\rangle | x \in X\} \tag{1}$$

Where $h_A(x)$ is the set of values in [0,1] representing the possible membership of the element $x$ with respect to set A.

### 3.2. The related concept of PHFS

Xu and Zhou [20] proposed PHFSs by introducing probabilistic information into HFSs to compensate for the loss of the preference information of DMs in HFSs.

**Definition 2**. [20] Let X be a fixed set: Then, the mathematical expression of PHFS A on X is

$$A = \left\{ \left\langle x, h_{Ax_i}\left(p_{Ax_i}\right)\right\rangle | x_i \in X\right\} \tag{2}$$

Here, $h_{Ax_i}\left(p_{Ax_i}\right)$ is composed of $\gamma_i|p_i$, $\gamma_i|p_i$ represents the fuzzy information with probability to set A, and $h_{Ax_i}\left(p_{Ax_i}\right)$ is called a PHFE by Xu and Zhou [20].

Where $\gamma_i$ satisfies $0 \leq \gamma_i \leq 1$, $p_i$ satisfies $0 \leq p_i \leq 1$ and $\sum_{i=1}^{l_h} p_i = 1$, $i = 1, 2, \cdots, l_h$, Here, $l_h$ represents the number of possible elements in the $h_{Ax_i}\left(p_{Ax_i}\right)$.

For the sake of convenience, in this study, $h_{Ax_i}\left(p_{Ax_i}\right)$ can be shortened as $h(p)$.

To compare PHFEs, Xu and Zhou [20] proposed the following comparison method:

**Definition 3**. [20] For PHFE $h(p)$, the scoring function of $h(p)$ can be expressed as:

$$s(h(p)) = \sum_{i=1}^{l_h} \gamma_i p_i \tag{3}$$

where $l_h$ represents the number of possible elements in $h(p)$.

We note that the larger $s(h(p))$ is, the better $h(p)$ is. However, there are some situations where the above sorting approach will be ineffective, such as:

**Example 1**. Let $h_1(p) = \{0.2|0.6, 0.8|0.4\}$ and $h_2(p) = \{0.6|0.2, 0.4|0.8\}$ be two PHFEs on A, then, $s(h_1(p)) = 0.2×0.6+0.8×0.4 = 0.44$, $s(h_2(p)) = 0.6×0.2+0.4×0.4 = 0.44$.

Clearly, in $s(h_1(p)) = s(h_2(p))$, we cannot compare $h_1(p)$ and $h_2(p)$ according to the scoring function. Therefore, considering this situation, Xu and Zhou [20] proposed a deviation function to compare $h_1(p)$ and $h_2(p)$ better. The deviation function of $h(p)$ can be expressed as

$$\delta(h(p)) = \sum_{i=1}^{l_h} (\gamma_i - s(h(p)))^2 p_i \tag{4}$$

Thus, the comparison rules for $h_1(p)$ and $h_2(p)$ are as follows:

If $s(h_1(p)) > s(h_2(p))$, then $h_1(p) > h_2(p)$;

If $s(h_1(p)) = s(h_2(p))$, then

1. if $\delta(h_1(p)) > \delta(h_2(p))$, then $h_1(p) < h_2(p)$;

2. if $\delta(h_1(p)) < \delta(h_2(p))$, then $h_1(p) > h_2(p)$;

3. if $\delta(h_1(p)) = \delta(h_2(p))$, then $h_1(p) = h_2(p)$.

Using the above rules, we obtain the following for the situation in Example 1:

$$\delta(h_1(p)) = (0.2 - 0.44)^2 \times 0.6 + (0.8 - 0.44)^2 \times 0.4 = 0.0864$$

$$\delta(h_2(p)) = (0.6 - 0.44)^2 \times 0.2 + (0.4 - 0.44)^2 \times 0.8 = 0.0064$$

Obviously, $\delta(h_1(p)) > \delta(h_2(p))$. Therefore, based on the above judgment principle, we can easily obtain the $h_1(p) < h_2(p)$.

## 3.3 Correlation coefficient for HFS

Referring to the practices of Gerstenkorn and Manko [40] and Bustince and Burillo [41] for intuitionistic fuzzy sets, Chen et al. [45] introduced the information energy proposed by Dumitrescu [56] to HFSs.

**Definition 4**. [45] Let A be the HFS. The information energy of A can then be defined as:

$$\vartheta(A) = \sum_{i=1}^{n} \left( \frac{1}{l_{x_i}} \sum_{j=1}^{l_{x_i}} \left( h_A^{\delta(j)}(x_i) \right)^2 \right) \tag{5}$$

Subsequently, Chen et al. [45] proposed a correlation between two HFSs based on Definition 4, which is defined as follows:

**Definition 5**. [45] Let $A_1$ and $A_2$ be two HFSs, then the correlation between them is:

$$C(A_1, A_2) = \sum_{i=1}^{n} \left( \frac{1}{l_{x_i}} \sum_{j=1}^{l_{x_i}} \left( h_{A_1}^{\delta(j)}(x_i) h_{A_2}^{\delta(j)}(x_i) \right) \right) \tag{6}$$

Using Definitions 4 and 5, Chen et al. [45] further proposed the correlation coefficient between the two HFSs as follows:

**Definition 6**. [45] Let $A_1$ and $A_2$ be two HFSs; then, the correlation coefficient between them is:

$$\rho(A_1, A_2) = \frac{C(A_1, A_2)}{C(A_1, A_1)^{1/2} \cdot C(A_2, A_2)^{1/2}} \tag{7}$$

Subsequently, Chen et al. [45] discussed the properties of the correlation coefficients in Definition 6 and obtained the following properties.

**Theorem 1**. [45] Let $A_1$ and $A_2$ be two HFSs; then, the correlation coefficient between them satisfies the following properties:

1. $\rho(A_1, A_2) = \rho(A_2, A_1)$;

2. $0 \le \rho(A_1, A_2) \le 1$;

3. $\rho(A_1, A_2) = 1$, if $A_1 = A_2$.

### 3.4 Correlation coefficient for PHFS

To investigate the correlation coefficient of the PHFS, Song et al. [54] introduced the idea of the mean and variance in statistics into the correlation of the PHFS and defined the covariance and mean of the PHFS. The mean and variance of the PHFS are defined as follows.

**Definition 7**. [54]: Let A be a PHFS, where $A = \left\{ \left\langle x, h_{Ax_i}\left(p_{Ax_i}\right) \right\rangle | x_i \in X \right\}$. The mean and variance of A are expressed as

$$\overline{A} = \frac{1}{n} \sum_{i=1}^{n} \bar{h}_{Ax_i}\left(p_{Ax_i}\right) \tag{8}$$

$$Var(A) = \frac{1}{n} \sum_{i=1}^{n} \left(\bar{h}_{Ax_i}\left(p_{Ax_i}\right) - \overline{A}\right) \tag{9}$$

Here, $\bar{h}_{Ax_i}\left(p_{Ax_i}\right) = \sum_{j=1}^{l_{x_i}} \left(\gamma_{Ax_i}{}^{\delta(j)} \cdot p_{Ax_i}{}^{\delta(j)}\right)$.

Based on the mean and covariance of the PHFS, Song et al. [54] defined the covariance and correlation coefficient between the two PHFSs as follows:

**Definition 8**. [54]: Let A and B be two PHFSs on universe X, where $A = \left\{ \left\langle x, h_{Ax_i}\left(p_{Ax_i}\right) \right\rangle | x_i \in X \right\}$ and $B = \left\{ \left\langle x, h_{Bx_i}\left(p_{Bx_i}\right) \right\rangle | x_i \in X \right\}$, then the covariance and correlation coefficient between A and B are, respectively, expressed as follows:

$$C(A, B) = \frac{1}{n} \sum_{i=1}^{n} \left[\bar{h}_{Ax_i}\left(p_{Ax_i}\right) - \overline{A}\right] \cdot \left[\bar{h}_{Bx_i}\left(p_{Bx_i}\right) - \overline{B}\right] \tag{10}$$

$$\rho(A, B) = \frac{C(A, B)}{C(A, A)^{1/2} \cdot C(B, B)^{1/2}} \tag{11}$$

Where $C(A, B)$ represents the covariance between A and B and $\rho(A, B)$ represents the correlation coefficient between A and B. According to the above formula, Song et al. [54] discussed and obtained some properties of the correlation coefficient, as follows:

**Theorem 2**. [54] Let A and B be the two PHFs. The correlation coefficients between them satisfy the following properties.

1. $\rho(A, B) = \rho(B, A)$;

2. $\rho(A, A) = 1$;

3. $\rho(A, A^c) = -1$;

4. $-1 \le \rho(A, B) \le 1$.

Subsequently, Song et al. [54] considered the weight and further proposed the weighted covariance and correlation coefficient between A and B, as follows:

$$C_w(A, B) = \frac{1}{n} \sum_{i=1}^{n} \left[ w_i \bar{h}_{Ax_i}\left(p_{Ax_i}\right) - \overline{A_w} \right] \cdot \left[ w_i \bar{h}_{Bx_i}\left(p_{Bx_i}\right) - \overline{B_w} \right] \qquad (12)$$

$$\rho_w(A, B) = \frac{C_w(A, B)}{C_w(A, A)^{1/2} \cdot C_w(B, B)^{1/2}} \qquad (13)$$

This correlation coefficient is widely used in MCDM and cluster analyses. Although they exhibit good properties, they also exhibit shortcomings. This is illustrated using an example.

**Example 2**. Assumes that there are three PHFSs, which are as follows:

$$A = \left\{ \left( 0.3|\frac{1}{2}, 0.5|\frac{1}{2} \right), \left( 0.3|\frac{1}{3}, 0.6|\frac{1}{3}, 0.9|\frac{1}{3} \right), \left( 0.1|\frac{1}{4}, 0.2|\frac{1}{4}, 0.8|\frac{1}{4}, 0.9|\frac{1}{4} \right) \right\}$$

$$B = \left\{ \left( 0.1|\frac{1}{2}, 0.7|\frac{1}{2} \right), \left( 0.2|\frac{1}{3}, 0.7|\frac{1}{3}, 0.9|\frac{1}{3} \right), \left( 0.3|\frac{1}{3}, 0.5|\frac{1}{3}, 0.7|\frac{1}{3} \right) \right\}$$

$$C = \left\{ \left( 0.2|\frac{1}{3}, 0.3|\frac{1}{3}, 0.7|\frac{1}{3} \right), \left( 0.5|\frac{1}{2}, 0.7|\frac{1}{2} \right), \left( 0.4|\frac{1}{3}, 0.5|\frac{1}{3}, 0.6|\frac{1}{3} \right) \right\}$$

According to formula (11) in Definition 8, we can calculate $\rho(A, B) = \rho(A, C) = \rho(B, C) = 1$, obviously $A \neq B \neq C$, which is contradictory to (2) in Theorem 2 proposed by Song et al. [54]. This is mainly because the means of the PHFEs corresponding to each other in A, B, and C are equal, which makes the mean value of PHFSs equal to each other, that is, $\overline{A} = \overline{B} = \overline{C}$, and then makes the variance of PHFSs equal to each other, that is, $Var(A) = Var(B) = Var(c)$. Finally, the correlation coefficients of the PHFSs are equal to each other, that is, $\rho(A, B) = \rho(A, C) = \rho(B, C) = 1$. If the mean value of each PHFE between multiple PHFSs is equal, it can be concluded that the correlation coefficient between multiple PHFSs is equal to 1. In addition, there is no evidence of a linear relationship between A,B, and C, so we cannot obtain a linear correlation between A,B, and C. In view of the above shortcomings, Liu and Guan [55] studied this problem and proposed a hybrid correlation coefficient for PHFSs, as follows:

**Definition 9**. [55] Let A and B be two PHFSs; then, the mixed correlation coefficient between them is

$$\rho_{MVL}(A, B) = \alpha \rho_M(A, B) + \beta \rho_V(A, B) + \lambda \rho_L(A, B) \qquad (14)$$

Where $\rho_{MVL}(A, B)$, $\rho_M(A, B)$, $\rho_V(A, B)$, and $\rho_L(A, B)$ represent the mixed correlation coefficient, mean correlation coefficient, variance correlation coefficient, and length rate correlation coefficient, respectively. In addition, $\alpha$, $\beta$, and $\lambda$ are the weights of the mean, variance, and length rate correlation coefficients, respectively, which satisfy $\alpha + \beta + \lambda = 1$. For specific expressions of $\rho_{MVL}(A, B)$, $\rho_M(A, B)$, $\rho_V(A, B)$, and $\rho_L(A, B)$, please refer to the literature [55].

We discovered that the method proposed by Liu and Guan [55] for calculating the correlation coefficient(14) is relatively complex and requires extensive calculations. Second, the correlation coefficient is dependent on the weight setting of the mean, variance, and length rate correlation coefficients, which makes the calculation of the correlation coefficient somewhat subjective. When the correlation coefficient is applied to a decision, the resulting decision may also be subjective.

Therefore, it is necessary to conduct additional research on the correlation coefficient of PHFS to accommodate more situations. This will be discussed in the following section.

## 4. Novel correlation coefficients for PHFS

In this section, we introduce the information energy proposed by Dumitrescu [56] into the PHFS, referring to the ideas of Gerstenkorn and Manko [40] and Chen et al. [45] for intuitionistic fuzzy sets and HFSs, respectively. On this basis, new correlation coefficients of the PHFS are proposed, and their properties are discussed and proved.

We first define the information energy of PHFSs as follows:

**Definition 10**. Let $A = \left\{ \left\langle x, h_{Ax_i}\left(p_{Ax_i}\right) \right\rangle | x_i \in X \right\}$ and $i = 1, 2, \cdots, n$ be a PHFS; then, its information energy can be

$$\psi(A) = \sum_{i=1}^{n} \left( \frac{1}{l_{x_i}} \sum_{j=1}^{l_{x_i}} \left( \gamma_{Ax_i}{}^{\delta(j)} \cdot p_{Ax_i}{}^{\delta(j)} \right)^2 \right) \tag{15}$$

Here, $l_{x_i}$ is the number of numerical values contained in PHFE corresponding to $X$ in $x_i$ in PHFS A, and $\gamma_{Ax_i}{}^{\delta(j)} \cdot p_{Ax_i}{}^{\delta(j)}$ represents the product of membership and probability corresponding to the j-th element in PHFE.

Based on Definition 10, we propose a correlation between two PHFSs as follows:

**Definition 11**. Let $A_1$ and $A_2$ be two PHFSs. Then, the correlation between them is

$$C_1(A_1, A_2) = \sum_{i=1}^{n} \left( \frac{1}{l_{x_i}} \sum_{j=1}^{l_{x_i}} \left( \gamma_{A_1 x_i}{}^{\delta(j)} \cdot \gamma_{A_2 x_i}{}^{\delta(j)} \cdot p_{A_1 x_i}{}^{\delta(j)} \cdot p_{A_2 x_i}{}^{\delta(j)} \right) \right) \tag{16}$$

According to Eq (16), for any $A_1$, $A_2$, the correlation satisfies the following properties:

1. $C_1(A_1, A_1) = \psi(A_1)$;

2. $C_1(A_1, A_2) = C_1(A_2, A_1)$.

According to Definitions 10 and 11, we propose a new correlation coefficient without considering the weight as follows:

**Definition 12**. Let $A_1$ and $A_2$ be two PHFSs; then, the correlation coefficient $\rho_1(A_1, A_2)$ between them is

$$
\begin{aligned}
\rho_1(A_1, A_2) &= \frac{C_1(A_1, A_2)}{C_1(A_1, A_1)^{1/2} \cdot C_1(A_2, A_2)^{1/2}} \\
&= \frac{\sum_{i=1}^{n} \left( \frac{1}{l_{x_i}} \sum_{j=1}^{l_{x_i}} \left( \gamma_{A_1 x_i}{}^{\delta(j)} \cdot \gamma_{A_2 x_i}{}^{\delta(j)} \cdot p_{A_1 x_i}{}^{\delta(j)} \cdot p_{A_2 x_i}{}^{\delta(j)} \right) \right)}{\left( \sum_{i=1}^{n} \left( \frac{1}{l_{x_i}} \sum_{j=1}^{l_{x_i}} \left( \gamma_{A_1 x_i}{}^{\delta(j)} \cdot p_{A_1 x_i}{}^{\delta(j)} \right)^2 \right) \right)^{1/2} \cdot \left( \sum_{i=1}^{n} \left( \frac{1}{l_{x_i}} \sum_{j=1}^{l_{x_i}} \left( \gamma_{A_2 x_i}{}^{\delta(j)} \cdot p_{A_2 x_i}{}^{\delta(j)} \right)^2 \right) \right)^{1/2}}
\end{aligned} \tag{17}
$$

**Note 1**: The numbers of values in different PHFEs are usually different, and these values are usually unordered. For convenience, we arrange the values in PHFEs in increasing order, satisfying $F^{\sigma(i)} \leq F^{\sigma(i+1)}$, where $F^{\sigma(i)} = p^{\sigma(i)} \gamma^{\sigma(i)}$ for a PHFE represents the i-th maximum value in the PHFE. Second, in order to calculate the correlation coefficient of two PHFSs, let $l_{x_i} = \max\left\{ l\left(h_{Ax_i}\left(p_{Ax_i}\right)\right), l\left(h_{Bx_i}\left(p_{Bx_i}\right)\right) \right\}$, when $l\left(h_{Ax_i}\left(p_{Ax_i}\right)\right) \neq l\left(h_{Bx_i}\left(p_{Bx_i}\right)\right)$, we need to add some elements to the PHFE with fewer elements according to the optimistic or pessimistic

criterion so that they have the same length. In this study, we adopt the pessimistic criterion: when $l\left(h_{Ax_i}\left(p_{Ax_i}\right)\right) \leq l\left(h_{Bx_i}\left(p_{Bx_i}\right)\right)$, we need to add the minimum value to $h_{Ax_i}\left(p_{Ax_i}\right)$ such that $h_{Ax_i}\left(p_{Ax_i}\right)$ and $h_{Bx_i}\left(p_{Bx_i}\right)$ have the same length.

**Example 2**.

Let, A = {(0.5|0.4, 0.8|0.3, 0.7|0.3), (0.1|0.5, 0.3|0.3, 0.6|0.2), (0.2|0.7, 0.3|0.3)} B = {(0.4|0.3, 0.5|0.7), (0.7|0.4, 0.3|0.1, 0.5|0.5), (0.3|0.2, 0.8|0.1, 0.7|0.7)} be two PHFSs, then the correlation coefficient between them is calculated as follows:

**Step 1**: Sort the values in the PHFEs in A and B and then supplement the probabilistic fuzzy elements with less length according to the pessimistic criterion. The above A and B can be transformed into

$$A^{\Delta} = \{(0.5|0.4, 0.7|0.3, 0.8|0.3), (0.1|0.5, 0.3|0.3, 0.6|0.2), (0.3|0, 0.3|0.3, 0.2|0.7)\}$$

$$B^{\Delta} = \{(0.4|0, 0.4|0.3, 0.5|0.7), (0.3|0.1, 0.2|0.4, 0.5|0.5), (0.3|0.2, 0.8|0.1, 0.7|0.7)\}$$

**Step 2**: The correlation coefficient between A and B is calculated using Eq (17).

$$C_1(A, A) = \frac{1}{3} \times \left[(0.5 \times 0.4)^2 + (0.8 \times 0.3)^2 + (0.7 \times 0.3)^2\right] + \frac{1}{3} \times \left[(0.1 \times 0.5)^2 + (0.3 \times 0.3)^2 + (0.6 \times 0.2)^2\right]$$
$$+ \frac{1}{2} \times \left[(0.2 \times 0.7)^2 + (0.3 \times 0.3)^2\right]$$
$$= 0.0694$$

$$C_1(B, B) = \frac{1}{2} \times \left[(0.4 \times 0.3)^2 + (0.5 \times 0.7)^2\right] + \frac{1}{3} \times \left[(0.2 \times 0.4)^2 + (0.3 \times 0.1)^2 + (0.5 \times 0.5)^2\right]$$
$$+ \frac{1}{3} \times \left[(0.3 \times 0.2)^2 + (0.8 \times 0.1)^2 + (0.7 \times 0.7)^2\right]$$
$$= 0.1751$$

$$C_1(A, B) = C_1(A^{\Delta}, B^{\Delta}) = \frac{1}{3} \times \left[(0.5 \times 0.4 \times 0.4 \times 0) + (0.7 \times 0.3 \times 0.4 \times 0.3) + (0.8 \times 0.3 \times 0.5 \times 0.7)\right]$$
$$+ \frac{1}{3} \times \left[(0.1 \times 0.5 \times 0.3 \times 0.1) + (0.3 \times 0.3 \times 0.2 \times 0.4) + (0.6 \times 0.2 \times 0.5 \times 0.5)\right]$$
$$+ \frac{1}{3} \times \left[(0.3 \times 0 \times 0.3 \times 0.2) + (0.3 \times 0.3 \times 0.8 \times 0.1) + (0.2 \times 0.7 \times 0.7 \times 0.7)\right]$$
$$= 0.0746$$

$$\rho_1(A, B) = \frac{C_1(A, B)}{C_1(A, A)^{1/2} \cdot C_1(B, B)^{1/2}}$$
$$= \frac{0.0746}{0.0694^{1/2} \times 0.1751^{1/2}} = 0.6764$$

Next, based on the newly proposed correlation coefficient measure in Definition 12, we obtain the new correlation coefficient formula (17) that satisfies the following properties:

**Theorem 3**. Let $A_1$ and $A_2$ be two PHFSs; then, the correlation coefficients between them satisfy the following properties:

1. $\rho_1(A_1, A_2) = \rho_1(A_2, A_1)$;

2. $0 \leq \rho_1(A_1, A_2) \leq 1$;

3. $\rho_1(A_1, A_2) = 1$, if $A_1 = A_2$.

Eq (17) clearly satisfies (1) and (3) in Theorem 3. Next, we prove that Eq (17) satisfies Eq (2) in Theorem 3.

**Proof.**

Obviously, $\rho_1(A_1, A_2) \geq 0$. Next, we prove $\rho_1(A_1, A_2) \leq 1$.

Since,

$$
\begin{aligned}
C_1(A_1, A_2) \quad &= \sum_{i=1}^{n} \left( \frac{1}{l_{x_i}} \sum_{j=1}^{l_{x_i}} \left( \gamma_{A_1 x_i}{}^{\delta(j)} \cdot \gamma_{A_2 x_i}{}^{\delta(j)} \cdot p_{A_1 x_i}{}^{\delta(j)} \cdot p_{A_2 x_i}{}^{\delta(j)} \right) \right) \\
&= \frac{1}{l_{x_1}} \sum_{j=1}^{l_{x_1}} \left( \gamma_{A_1 x_1}{}^{\delta(j)} \cdot \gamma_{A_2 x_1}{}^{\delta(j)} \cdot p_{A_1 x_1}{}^{\delta(j)} \cdot p_{A_2 x_1}{}^{\delta(j)} \right) + \frac{1}{l_{x_2}} \sum_{j=1}^{l_{x_2}} \left( \gamma_{A_1 x_2}{}^{\delta(j)} \cdot \gamma_{A_2 x_2}{}^{\delta(j)} \cdot p_{A_1 x_2}{}^{\delta(j)} \cdot p_{A_2 x_2}{}^{\delta(j)} \right) \\
&\quad + \cdots + \frac{1}{l_{x_n}} \sum_{j=1}^{l_{x_n}} \left( \gamma_{A_1 x_n}{}^{\delta(j)} \cdot \gamma_{A_2 x_n}{}^{\delta(j)} \cdot p_{A_1 x_n}{}^{\delta(j)} \cdot p_{A_2 x_n}{}^{\delta(j)} \right) \\
&= \sum_{j=1}^{l_{x_1}} \frac{\gamma_{A_1 x_1}{}^{\delta(j)} \cdot p_{A_1 x_1}{}^{\delta(j)}}{\sqrt{l_{x_1}}} \cdot \frac{\gamma_{A_2 x_1}{}^{\delta(j)} \cdot p_{A_2 x_1}{}^{\delta(j)}}{\sqrt{l_{x_1}}} + \sum_{j=1}^{l_{x_2}} \frac{\gamma_{A_1 x_2}{}^{\delta(j)} \cdot p_{A_1 x_2}{}^{\delta(j)}}{\sqrt{l_{x_2}}} \cdot \frac{\gamma_{A_2 x_2}{}^{\delta(j)} \cdot p_{A_2 x_2}{}^{\delta(j)}}{\sqrt{l_{x_2}}} \\
&\quad + \cdots + \sum_{j=1}^{l_{x_n}} \frac{\gamma_{A_1 x_n}{}^{\delta(j)} \cdot p_{A_1 x_1}{}^{\delta(j)}}{\sqrt{l_{x_n}}} \cdot \frac{\gamma_{A_2 x_n}{}^{\delta(j)} \cdot p_{A_2 x_n}{}^{\delta(j)}}{\sqrt{l_{x_n}}}
\end{aligned}
$$

Using the Cauchy—Schwarz inequality, namely
$(a_1 b_1 + a_2 b_2 + \cdots + a_n b_n)^2 \leq \left( a_1^2 + a_2^2 + \cdots + a_n^2 \right) \cdot \left( b_1^2 + b_2^2 + \cdots + b_n^2 \right)$, then we get:

$$
\begin{aligned}
(C_1(A_1, A_2))^2 \quad &\leq \left[ \sum_{j=1}^{l_{x_1}} \frac{\left( \gamma_{A_1 x_1}{}^{\delta(j)} \cdot p_{A_1 x_1}{}^{\delta(j)} \right)^2}{l_{x_1}} + \sum_{j=1}^{l_{x_2}} \frac{\left( \gamma_{A_1 x_2}{}^{\delta(j)} \cdot p_{A_1 x_2}{}^{\delta(j)} \right)^2}{l_{x_2}} + \cdots + \sum_{j=1}^{l_{x_n}} \frac{\left( \gamma_{A_1 x_n}{}^{\delta(j)} \cdot p_{A_1 x_n}{}^{\delta(j)} \right)^2}{l_{x_n}} \right] \\
&\quad \times \left[ \sum_{j=1}^{l_{x_1}} \frac{\left( \gamma_{A_2 x_1}{}^{\delta(j)} \cdot p_{A_2 x_1}{}^{\delta(j)} \right)^2}{l_{x_1}} + \sum_{j=1}^{l_{x_2}} \frac{\left( \gamma_{A_2 x_2}{}^{\delta(j)} \cdot p_{A_2 x_2}{}^{\delta(j)} \right)^2}{l_{x_2}} + \cdots + \sum_{j=1}^{l_{x_n}} \frac{\left( \gamma_{A_2 x_n}{}^{\delta(j)} \cdot p_{A_2 x_n}{}^{\delta(j)} \right)^2}{l_{x_n}} \right] \\
&= \left[ \frac{1}{l_{x_1}} \sum_{j=1}^{l_{x_1}} \left( \gamma_{A_1 x_1}{}^{\delta(j)} \cdot p_{A_1 x_1}{}^{\delta(j)} \right)^2 + \frac{1}{l_{x_2}} \sum_{j=1}^{l_{x_2}} \left( \gamma_{A_1 x_2}{}^{\delta(j)} \cdot p_{A_1 x_2}{}^{\delta(j)} \right)^2 + \cdots + \frac{1}{l_{x_n}} \sum_{j=1}^{l_{x_n}} \left( \gamma_{A_1 x_n}{}^{\delta(j)} \cdot p_{A_1 x_n}{}^{\delta(j)} \right)^2 \right] \\
&\quad \times \left[ \frac{1}{l_{x_1}} \sum_{j=1}^{l_{x_1}} \left( \gamma_{A_2 x_1}{}^{\delta(j)} \cdot p_{A_2 x_1}{}^{\delta(j)} \right)^2 + \frac{1}{l_{x_2}} \sum_{j=1}^{l_{x_2}} \left( \gamma_{A_2 x_2}{}^{\delta(j)} \cdot p_{A_2 x_2}{}^{\delta(j)} \right)^2 + \cdots + \frac{1}{l_{x_n}} \sum_{j=1}^{l_{x_n}} \left( \gamma_{A_2 x_n}{}^{\delta(j)} \cdot p_{A_2 x_n}{}^{\delta(j)} \right)^2 \right] \\
&= \left[ \sum_{i=1}^{n} \frac{1}{l_{x_i}} \sum_{j=1}^{l_{x_i}} \left( \gamma_{A_1 x_1}{}^{\delta(j)} \cdot p_{A_1 x_1}{}^{\delta(j)} \right)^2 \right] \times \left[ \sum_{i=1}^{n} \frac{1}{l_{x_i}} \sum_{j=1}^{l_{x_i}} \left( \gamma_{A_2 x_1}{}^{\delta(j)} \cdot p_{A_2 x_1}{}^{\delta(j)} \right)^2 \right] \\
&= C_1(A_1, A_1) \times C_1(A_2, A_2)
\end{aligned}
$$

Therefore, we can further obtain,

$$C_1(A_1, A_2) \le (C_1(A_1, A_1))^{1/2} \times (C_1(A_2, A_2))^{1/2}$$

$$\Rightarrow \quad \frac{C_1(A_1, A_2)}{(C_1(A_1, A_1))^{1/2} \times (C_1(A_2, A_2))^{1/2}} \le 1$$

$$\Rightarrow \quad \rho_1(A_1, A_2) \le 1$$

Therefore,

$$0 \le \rho_1(A_1, A_2) \le 1$$

Next, according to the newly proposed formula (17), we calculated the correlation coefficient between the three PHFSs in Example 1:

$$
\begin{aligned}
C_1(A, A) \quad &= \frac{1}{2} \times \left[ \left(0.3 \times \frac{1}{2}\right)^2 + \left(0.5 \times \frac{1}{2}\right)^2 \right] + \frac{1}{3} \times \left[ \left(0.3 \times \frac{1}{3}\right)^2 + \left(0.6 \times \frac{1}{3}\right)^2 + \left(0.9 \times \frac{1}{3}\right)^2 \right] \\
&\quad + \frac{1}{4} \times \left[ \left(0.1 \times \frac{1}{4}\right)^2 + \left(0.2 \times \frac{1}{4}\right)^2 + \left(0.8 \times \frac{1}{4}\right)^2 + \left(0.9 \times \frac{1}{4}\right)^2 \right] \\
&= 0.1126
\end{aligned}
$$

$$
\begin{aligned}
C_1(B, B) \quad &= \frac{1}{2} \times \left[ \left(0.1 \times \frac{1}{2}\right)^2 + \left(0.7 \times \frac{1}{2}\right)^2 \right] + \frac{1}{3} \times \left[ \left(0.2 \times \frac{1}{3}\right)^2 + \left(0.7 \times \frac{1}{3}\right)^2 + \left(0.9 \times \frac{1}{3}\right)^2 \right] \\
&\quad + \frac{1}{3} \times \left[ \left(0.3 \times \frac{1}{3}\right)^2 + \left(0.5 \times \frac{1}{3}\right)^2 + \left(0.7 \times \frac{1}{3}\right)^2 \right] \\
&= 0.1429
\end{aligned}
$$

$$
\begin{aligned}
C_1(C, C) \quad &= \frac{1}{3} \times \left[ \left(0.2 \times \frac{1}{3}\right)^2 + \left(0.3 \times \frac{1}{3}\right)^2 + \left(0.7 \times \frac{1}{3}\right)^2 \right] + \frac{1}{2} \times \left[ \left(0.5 \times \frac{1}{2}\right)^2 + \left(0.7 \times \frac{1}{2}\right)^2 \right] \\
&\quad + \frac{1}{3} \times \left[ \left(0.4 \times \frac{1}{3}\right)^2 + \left(0.5 \times \frac{1}{3}\right)^2 + \left(0.6 \times \frac{1}{3}\right)^2 \right] \\
&= 0.1440
\end{aligned}
$$

$$
\begin{aligned}
C_1(A, B) \quad &= \frac{1}{2} \times \left[ \left(0.3 \times \frac{1}{2} \times 0.1 \times \frac{1}{2}\right) + \left(0.5 \times \frac{1}{2} \times 0.7 \times \frac{1}{2}\right) \right] \\
&\quad + \frac{1}{3} \times \left[ \left(0.3 \times \frac{1}{3} \times 0.2 \times \frac{1}{3}\right) + \left(0.6 \times \frac{1}{3} \times 0.7 \times \frac{1}{3}\right) + \left(0.9 \times \frac{1}{3} \times 0.9 \times \frac{1}{3}\right) \right] \\
&\quad + \frac{1}{4} \times \left[ \left(0.1 \times \frac{1}{4} \times 0.3 \times 0\right) + \left(0.2 \times \frac{1}{4} \times 0.3 \times \frac{1}{3}\right) + \left(0.8 \times \frac{1}{4} \times 0.5 \times \frac{1}{3}\right) + \left(0.9 \times \frac{1}{4} \times 0.7 \times \frac{1}{3}\right) \right] \\
&= 0.1180
\end{aligned}
$$

$$
\begin{aligned}
C_1(A, C) \quad &= \frac{1}{3} \times \left[ \left( 0.3 \times 0 \times 0.2 \times \frac{1}{3} \right) + \left( 0.3 \times \frac{1}{2} \times 0.3 \times \frac{1}{3} \right) + \left( 0.5 \times \frac{1}{2} \times 0.7 \times \frac{1}{3} \right) \right] \\
&+ \frac{1}{3} \times \left[ \left( 0.3 \times \frac{1}{3} \times 0.5 \times 0 \right) + \left( 0.6 \times \frac{1}{3} \times 0.5 \times \frac{1}{2} \right) + \left( 0.9 \times \frac{1}{3} \times 0.7 \times \frac{1}{2} \right) \right] \\
&+ \frac{1}{4} \times \left[ \left( 0.1 \times \frac{1}{4} \times 0.4 \times 0 \right) + \left( 0.2 \times \frac{1}{4} \times 0.4 \times \frac{1}{3} \right) + \left( 0.8 \times \frac{1}{4} \times 0.5 \times \frac{1}{3} \right) + \left( 0.9 \times \frac{1}{4} \times 0.6 \times \frac{1}{3} \right) \right] \\
&= 0.0974
\end{aligned}
$$

$$
\begin{aligned}
C_1(B, C) \quad &= \frac{1}{3} \times \left[ \left( 0.1 \times 0 \times 0.2 \times \frac{1}{3} \right) + \left( 0.1 \times \frac{1}{2} \times 0.3 \times \frac{1}{3} \right) + \left( 0.7 \times \frac{1}{2} \times 0.7 \times \frac{1}{3} \right) \right] \\
&+ \frac{1}{3} \times \left[ \left( 0.2 \times \frac{1}{3} \times 0.5 \times 0 \right) + \left( 0.7 \times \frac{1}{3} \times 0.5 \times \frac{1}{2} \right) + \left( 0.9 \times \frac{1}{3} \times 0.7 \times \frac{1}{2} \right) \right] \\
&+ \frac{1}{3} \times \left[ \left( 0.3 \times \frac{1}{3} \times 0.4 \times \frac{1}{3} \right) + \left( 0.5 \times \frac{1}{3} \times 0.5 \times \frac{1}{3} \right) + \left( 0.7 \times \frac{1}{3} \times 0.6 \times \frac{1}{3} \right) \right] \\
&= 0.1126
\end{aligned}
$$

Then,

$$
\begin{aligned}
\rho_1(A, B) \quad &= \frac{C_1(A, B)}{C_1(A, A)^{1/2} \cdot C_1(B, B)^{1/2}} \\
&= \frac{0.1180}{0.1126^{1/2} \times 0.1429^{1/2}} = 0.9302
\end{aligned}
$$

$$
\begin{aligned}
\rho_1(A, C) \quad &= \frac{C_1(A, C)}{C_1(A, A)^{1/2} \cdot C_1(C, C)^{1/2}} \\
&= \frac{0.0974}{0.1126^{1/2} \times 0.1440^{1/2}} = 0.7646
\end{aligned}
$$

$$
\begin{aligned}
\rho_1(B, C) \quad &= \frac{C_1(B, C)}{C_1(B, B)^{1/2} \cdot C_1(C, C)^{1/2}} \\
&= \frac{0.1126}{0.1429^{1/2} \times 0.1440^{1/2}} = 0.7850
\end{aligned}
$$

Using our newly proposed correlation coefficient formula (17), we can obtain $\rho_1(A, B) > \rho_1(B, C) > \rho_1(A, C)$ and overcome the defects in formula (11), which means that our proposed correlation coefficient formula is effective.

Next, we extend Eq (17) and propose a new correlation coefficient formula as follows:

**Definition 13**. Let $A_1$ and $A_2$ be two PHFSs; then, the correlation coefficient $\rho_2(A_1, A_2)$ between them is

$$
\begin{aligned}
\rho_2(A_1, A_2) \quad &= \frac{C_1(A_1, A_2)}{\max\{C_1(A_1, A_1) \cdot C_1(A_2, A_2)\}} \\
&= \frac{\sum_{i=1}^{n}\left(\frac{1}{l_{x_i}}\sum_{j=1}^{l_{x_i}}\left(\gamma_{A_1x_i}{}^{\delta(j)} \cdot \gamma_{A_2x_i}{}^{\delta(j)} \cdot p_{A_1x_i}{}^{\delta(j)} \cdot p_{A_2x_i}{}^{\delta(j)}\right)\right)}{\max\left(\sum_{i=1}^{n}\left(\frac{1}{l_{x_i}}\sum_{j=1}^{l_{x_i}}\left(\gamma_{A_1x_i}{}^{\delta(j)} \cdot p_{A_1x_i}{}^{\delta(j)}\right)^2\right)\right) \cdot \left(\sum_{i=1}^{n}\left(\frac{1}{l_{x_i}}\sum_{j=1}^{l_{x_i}}\left(\gamma_{A_2x_i}{}^{\delta(j)} \cdot p_{A_2x_i}{}^{\delta(j)}\right)^2\right)\right)}
\end{aligned}
\tag{18}
$$

Eq (18) also satisfies the property in Theorem 2, which we prove as follows.

It is clear that formula (18) satisfies (1) and (3) in Theorem 3, and we only prove that formula (18) satisfies (2) in Theorem 3.

**Proof**.

Based on the proof of Eq (17) for Theorem (2), we obtain

$$
C_1(A_1, A_2) \leq (C_1(A_1, A_1))^{1/2} \times (C_1(A_2, A_2))^{1/2}
$$

Then we can further obtain,

$$
\frac{C_1(A_1, A_2)}{\max\{C_1(A_1, A_1), C_1(A_2, A_2)\}} \leq 1
$$

Therefore,

$$
0 \leq \rho_2(A_1, A_2) \leq 1.
$$

Next, we consider the weight and propose weighted correlation coefficients, as follows:

**Definition 14**. Let $A_1$ and $A_2$ be two PHFSs, and let the weighted correlation coefficients $\rho_3(A_1, A_2)$ and $\rho_4(A_1, A_2)$ be

$$
\begin{aligned}
\rho_3(A_1, A_2) \quad &= \frac{C_2(A_1, A_2)}{C_2(A_1, A_1)^{1/2} \cdot C_2(A_2, A_2)^{1/2}} \\
&= \frac{\sum_{i=1}^{n}w_i\left(\frac{1}{l_{x_i}}\sum_{j=1}^{l_{x_i}}\left(\gamma_{A_1x_i}{}^{\delta(j)} \cdot \gamma_{A_2x_i}{}^{\delta(j)} \cdot p_{A_1x_i}{}^{\delta(j)} \cdot p_{A_2x_i}{}^{\delta(j)}\right)\right)}{\left(\sum_{i=1}^{n}w_i\left(\frac{1}{l_{x_i}}\sum_{j=1}^{l_{x_i}}\left(\gamma_{A_1x_i}{}^{\delta(j)} \cdot p_{A_1x_i}{}^{\delta(j)}\right)^2\right)\right)^{1/2} \cdot \left(\sum_{i=1}^{n}w_i\left(\frac{1}{l_{x_i}}\sum_{j=1}^{l_{x_i}}\left(\gamma_{A_2x_i}{}^{\delta(j)} \cdot p_{A_2x_i}{}^{\delta(j)}\right)^2\right)\right)^{1/2}}
\end{aligned}
\tag{19}
$$

$$
\begin{aligned}
\rho_4(A_1, A_2) \quad &= \frac{C_2(A_1, A_2)}{\max\{C_2(A_1, A_1) \cdot C_2(A_2, A_2)\}} \\
&= \frac{\sum_{i=1}^{n}w_i\left(\frac{1}{l_{x_i}}\sum_{j=1}^{l_{x_i}}\left(\gamma_{A_1x_i}{}^{\delta(j)} \cdot \gamma_{A_2x_i}{}^{\delta(j)} \cdot p_{A_1x_i}{}^{\delta(j)} \cdot p_{A_2x_i}{}^{\delta(j)}\right)\right)}{\max\left(\sum_{i=1}^{n}w_i\left(\frac{1}{l_{x_i}}\sum_{j=1}^{l_{x_i}}\left(\gamma_{A_1x_i}{}^{\delta(j)} \cdot p_{A_1x_i}{}^{\delta(j)}\right)^2\right)\right) \cdot \left(\sum_{i=1}^{n}w_i\left(\frac{1}{l_{x_i}}\sum_{j=1}^{l_{x_i}}\left(\gamma_{A_2x_i}{}^{\delta(j)} \cdot p_{A_2x_i}{}^{\delta(j)}\right)^2\right)\right)}
\end{aligned}
\tag{20}
$$

According to the weighted correlation coefficients $\rho_3(A_1, A_2)$ and $\rho_4(A_1, A_2)$, we can see that if $w = \left(\frac{1}{n}, \frac{1}{n}, \cdots, \frac{1}{n}\right)^T$, then $\rho_3(A_1, A_2)$ and $\rho_4(A_1, A_2)$ degenerate into $\rho_1(A_1, A_2)$ and $\rho_2(A_1, A_2)$, respectively. Eqs (19) and (20) satisfy the following properties:

Theorem 4. If $A_1$ and $A_2$ are two PHFSs, $w_i(i = 1, 2, \cdots, n)$ is the weight of $x_i(i = 1, 2, \cdots, n)$ that satisfies $w_i \geq 0$ and $\sum_{i=1}^{n} w_i = 1$. The weighted correlation coefficient between the two satisfies the following properties:

1. $\rho_w(A_1, A_2) = \rho_w(A_2, A_1)$;

2. $0 \leq \rho_w(A_1, A_2) \leq 1$;

3. $\rho_w(A_1, A_2) = 1$, if $A_1 = A_2$.

Next, we prove that $\rho_3(A_1, A_2)$ satisfies Theorem 4, that $\rho_3(A_1, A_2)$ and $\rho_4(A_1, A_2)$ satisfy (1) and (3) in Theorem 3, and that $\rho_3(A_1, A_2)$ and $\rho_4(A_1, A_2)$ satisfy (2) in Theorem 4.

**Proof**.

Obviously, $\rho_3(A_1, A_2) \geq 0$. Next, we prove $\rho_3(A_1, A_2) \leq 1$.

Since,

$$
\begin{aligned}
C_2(A_1, A_2) &= \sum_{i=1}^{n} w_i \left( \frac{1}{l_{x_i}} \sum_{j=1}^{l_{x_i}} \left( \gamma_{A_1 x_i}{}^{\delta(j)} \cdot \gamma_{A_2 x_i}{}^{\delta(j)} \cdot p_{A_1 x_i}{}^{\delta(j)} \cdot p_{A_2 x_i}{}^{\delta(j)} \right) \right) \\
&= \frac{w_1}{l_{x_1}} \sum_{j=1}^{l_{x_1}} \left( \gamma_{A_1 x_1}{}^{\delta(j)} \cdot \gamma_{A_2 x_1}{}^{\delta(j)} \cdot p_{A_1 x_1}{}^{\delta(j)} \cdot p_{A_2 x_1}{}^{\delta(j)} \right) + \frac{w_2}{l_{x_2}} \sum_{j=1}^{l_{x_2}} \left( \gamma_{A_1 x_2}{}^{\delta(j)} \cdot \gamma_{A_2 x_2}{}^{\delta(j)} \cdot p_{A_1 x_2}{}^{\delta(j)} \cdot p_{A_2 x_2}{}^{\delta(j)} \right) \\
&\quad + \cdots + \frac{w_n}{l_{x_n}} \sum_{j=1}^{l_{x_n}} \left( \gamma_{A_1 x_n}{}^{\delta(j)} \cdot \gamma_{A_2 x_n}{}^{\delta(j)} \cdot p_{A_1 x_n}{}^{\delta(j)} \cdot p_{A_2 x_n}{}^{\delta(j)} \right) \\
&= \sum_{j=1}^{l_{x_1}} \frac{\sqrt{w_1} \cdot \gamma_{A_1 x_1}{}^{\delta(j)} \cdot p_{A_1 x_1}{}^{\delta(j)}}{\sqrt{l_{x_1}}} \cdot \frac{\sqrt{w_1} \cdot \gamma_{A_2 x_1}{}^{\delta(j)} \cdot p_{A_2 x_1}{}^{\delta(j)}}{\sqrt{l_{x_1}}} + \sum_{j=1}^{l_{x_2}} \frac{\sqrt{w_2} \cdot \gamma_{A_1 x_2}{}^{\delta(j)} \cdot p_{A_1 x_2}{}^{\delta(j)}}{\sqrt{l_{x_2}}} \cdot \frac{\sqrt{w_2} \cdot \gamma_{A_2 x_2}{}^{\delta(j)} \cdot p_{A_2 x_2}{}^{\delta(j)}}{\sqrt{l_{x_2}}} \\
&\quad + \cdots + \sum_{j=1}^{l_{x_n}} \frac{\sqrt{w_n} \cdot \gamma_{A_1 x_n}{}^{\delta(j)} \cdot p_{A_1 x_1}{}^{\delta(j)}}{\sqrt{l_{x_n}}} \cdot \frac{\sqrt{w_n} \cdot \gamma_{A_2 x_n}{}^{\delta(j)} \cdot p_{A_2 x_n}{}^{\delta(j)}}{\sqrt{l_{x_n}}}
\end{aligned}
$$

Using the Cauchy-Schwarz inequality, we get,

$$
(C_2(A_1,A_2))^2 \leq \left[ \sum_{j=1}^{l_{x_1}} \frac{w_1 \left( \gamma_{A_1 x_1}{}^{\delta(j)} \cdot p_{A_1 x_1}{}^{\delta(j)} \right)^2}{l_{x_1}} + \sum_{j=1}^{l_{x_2}} \frac{w_2 \left( \gamma_{A_1 x_2}{}^{\delta(j)} \cdot p_{A_1 x_2}{}^{\delta(j)} \right)^2}{l_{x_2}} + \cdots + \sum_{j=1}^{l_{x_n}} \frac{w_n \left( \gamma_{A_1 x_n}{}^{\delta(j)} \cdot p_{A_1 x_n}{}^{\delta(j)} \right)^2}{l_{x_n}} \right]
$$

$$
\times \left[ \sum_{j=1}^{l_{x_1}} \frac{w_1 \left( \gamma_{A_2 x_1}{}^{\delta(j)} \cdot p_{A_2 x_1}{}^{\delta(j)} \right)^2}{l_{x_1}} + \sum_{j=1}^{l_{x_2}} \frac{w_2 \left( \gamma_{A_2 x_2}{}^{\delta(j)} \cdot p_{A_2 x_2}{}^{\delta(j)} \right)^2}{l_{x_2}} + \cdots + \sum_{j=1}^{l_{x_n}} \frac{w_n \left( \gamma_{A_2 x_n}{}^{\delta(j)} \cdot p_{A_2 x_n}{}^{\delta(j)} \right)^2}{l_{x_n}} \right]
$$

$$
= \left[ \frac{w_1}{l_{x_1}} \sum_{j=1}^{l_{x_1}} \left( \gamma_{A_1 x_1}{}^{\delta(j)} \cdot p_{A_1 x_1}{}^{\delta(j)} \right)^2 + \frac{w_2}{l_{x_2}} \sum_{j=1}^{l_{x_2}} \left( \gamma_{A_1 x_2}{}^{\delta(j)} \cdot p_{A_1 x_2}{}^{\delta(j)} \right)^2 + \cdots + \frac{w_n}{l_{x_n}} \sum_{j=1}^{l_{x_n}} \left( \gamma_{A_1 x_n}{}^{\delta(j)} \cdot p_{A_1 x_n}{}^{\delta(j)} \right)^2 \right]
$$

$$
\times \left[ \frac{w_1}{l_{x_1}} \sum_{j=1}^{l_{x_1}} \left( \gamma_{A_2 x_1}{}^{\delta(j)} \cdot p_{A_2 x_1}{}^{\delta(j)} \right)^2 + \frac{w_2}{l_{x_2}} \sum_{j=1}^{l_{x_2}} \left( \gamma_{A_2 x_2}{}^{\delta(j)} \cdot p_{A_2 x_2}{}^{\delta(j)} \right)^2 + \cdots + \frac{w_n}{l_{x_n}} \sum_{j=1}^{l_{x_n}} \left( \gamma_{A_2 x_n}{}^{\delta(j)} \cdot p_{A_2 x_n}{}^{\delta(j)} \right)^2 \right]
$$

$$
= \left[ \sum_{i=1}^{n} \frac{w_i}{l_{x_i}} \sum_{j=1}^{l_{x_i}} \left( \gamma_{A_1 x_1}{}^{\delta(j)} \cdot p_{A_1 x_1}{}^{\delta(j)} \right)^2 \right] \times \left[ \sum_{i=1}^{n} \frac{w_i}{l_{x_i}} \sum_{j=1}^{l_{x_i}} \left( \gamma_{A_2 x_1}{}^{\delta(j)} \cdot p_{A_2 x_1}{}^{\delta(j)} \right)^2 \right]
$$

$$
= \left[ \sum_{i=1}^{n} w_i \left( \frac{1}{l_{x_i}} \sum_{j=1}^{l_{x_i}} \left( \gamma_{A_1 x_1}{}^{\delta(j)} \cdot p_{A_1 x_1}{}^{\delta(j)} \right)^2 \right) \right] \times \left[ \sum_{i=1}^{n} w_i \left( \frac{1}{l_{x_i}} \sum_{j=1}^{l_{x_i}} \left( \gamma_{A_2 x_1}{}^{\delta(j)} \cdot p_{A_2 x_1}{}^{\delta(j)} \right)^2 \right) \right]
$$

$$
= C_2(A_1,A_1) \times C_2(A_2,A_2)
$$

Therefore, we can further obtain,

$$
C_2(A_1,A_2) \leq (C_2(A_1,A_1))^{1/2} \times (C_2(A_2,A_2))^{1/2}
$$

$$
\Rightarrow \quad \frac{C_2(A_1,A_2)}{(C_2(A_1,A_1))^{1/2} \times (C_2(A_2,A_2))^{1/2}} \leq 1
$$

$$
\Rightarrow \quad \rho_3(A_1,A_2) \leq 1
$$

Therefore,

$$
0 \leq \rho_3(A_1,A_2) \leq 1.
$$

The proof of whether formula (20) satisfies Theorem 3 can be referred to the proof procedure of formula (18) for Theorem 2; therefore, we will not show the proof again.

## 5. A MCGDM method based on novel correlation coefficient of PHFSs under unknown weights

In this section, we propose a new MCGDM method based on the correlation coefficients of the PHFSs under unknown weights. In this method, we first refer to the practice of Chen and Xu [89] in HFS and propose a method to transform the evaluation information of the linguistic variables of DMs into PHF information. Subsequently, based on this method, we obtained the group decision matrix and weight of each evaluation criterion. Finally, we extended the new correlation coefficient and method to the MCGDM.

For an MCGDM problem, suppose $A = \{A_1, A_2 \cdots A_n\}$ is the set of alternatives, $C = \{C_1, C_2 \cdots C_m\}$ is the set of all criteria, where the criteria weights $w_j(1, 2, \cdots, m)$ meet $0 \leq w_j \leq 1, \sum_{j=1}^{m} w_j = 1$ and are independent of each other; $D = \{D_1, D_2 \cdots D_k\}$ represents the set of all

**Table 1. Translation between linguistic variables and Saaty's 1–9 scale.**

| Linguistic variables | Abbreviations | Saaty's 1–9 scale |
|---|---|---|
| Very low | VL | 1 |
| Very low to low | VLL | 2 |
| low | L | 3 |
| Medium low | ML | 4 |
| Medium | M | 5 |
| Medium high | MH | 6 |
| High | H | 7 |
| High to very high | HVH | 8 |
| Very high | VH | 9 |

DMs, where the weights $w_q(q = 1, 2\cdots, k)$ of DMs are the same, and any criterion $C_i$ and any alternative $A_i$ under $C_i$ are evaluated by each DM in terms of linguistic variables. Second, we refer to the practice of Chen and Xu [89] to transform the linguistic variable into Saaty's 1–9 scale, as detailed in Table 1. A decision flowchart is shown in Fig 1. The detailed steps of the decision-making method are as follows.

**Step 1**. Construct individual decision matrix for each DM.

Each DM evaluates all alternatives through linguistic variables to obtain the individual decision matrix of each DM.

**Step 2**. Evaluate the importance of each criterion.

Each DM assesses the importance of all the criteria involved in the alternative through language variables.

**Step 3**: Obtain the criteria weights.

According to Table 1, the evaluation values of the criteria given by the DMs were transformed according to expert weights to obtain the criteria weights. The specific process for calculating these weights is described in the next section.

**Step 4**: Construct individual weighted numerical decision matrix.

According to Saaty's 1-9 scale in Table 1, the individual decision matrix of each expert evaluated by linguistic variables is transformed into an individual weighted numerical decision matrix according to the weight of each expert. The specific transformation method is as follows. Suppose there are four expert $D = \{D_1, D_2\cdots D_4\}$ to evaluate the alternative $A_1$ under the criterion $C_1$ by linguistic variables, where each expert has equal weight. Therefore, we obtained the following decision results, as listed in Table 2.

Then the weighted evaluation values of the first and second experts can be expressed by $h_{A_1C_1}{}^{D_1} = 9 \times 0.25 = 2.25$ and $h_{A_1C_1}{}^{D_2} = 9 \times 0.25 = 2.25$, respectively, and the weighted evaluation values of the third and fourth experts can be denoted by $h_{A_1C_1}{}^{D_3} = 8 \times 0.25 = 2$ and $h_{A_1C_1}{}^{D_4} = 8 \times 0.25 = 2$, respectively.

**Step 5**: Construct hesitant fuzzy group decision matrix.

The individual weighted numerical decision matrix of each expert in step 4 is integrated into the hesitant fuzzy group decision matrix.

**Step 6**: Standardize hesitant fuzzy group decision matrix.

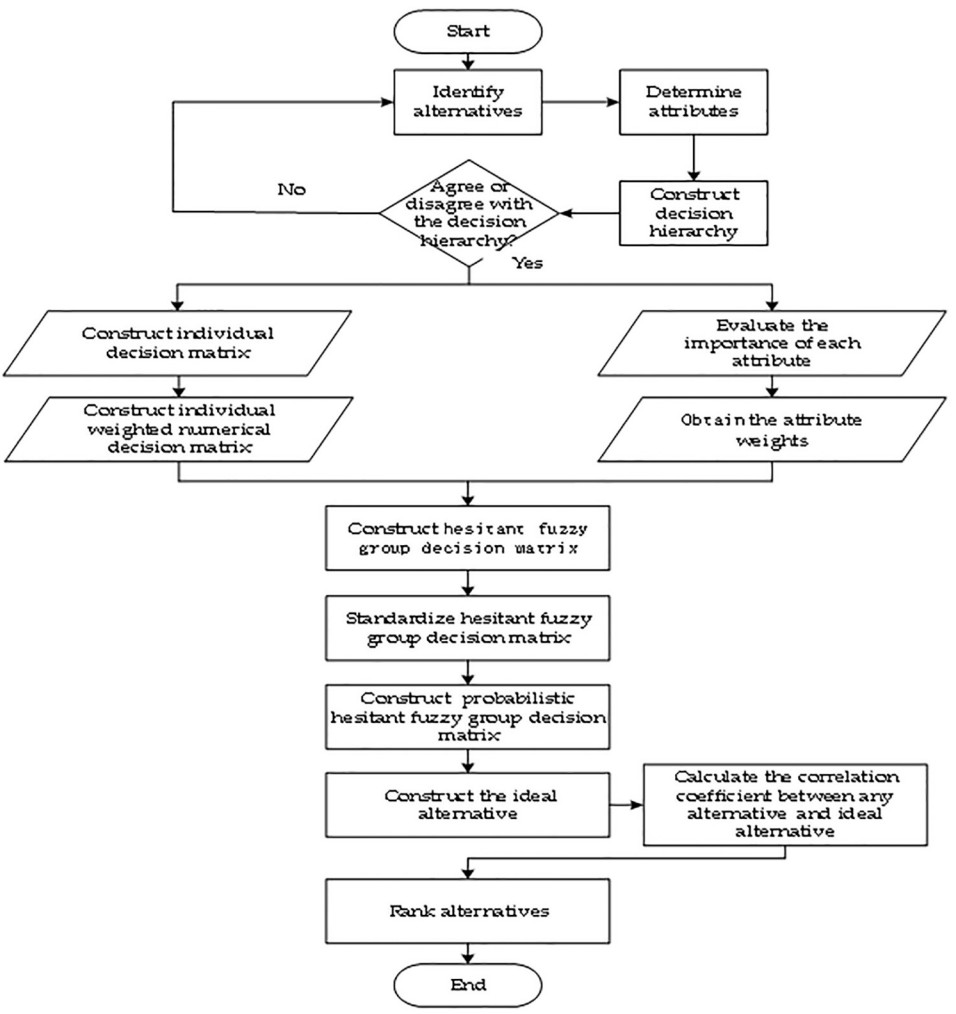

**Fig 1. Decision flow chart.**

We referred to the practice of Chen and Xu [89] to standardize the hesitant fuzzy group decision matrix, as follows:

$$
\overset{+}{h}_{ij}^{\,f} = \frac{h_{ij}^{\,f}}{\max\limits_{1 \le i \le k}\left\{\max\left\{h_{ij}^{\,1}, h_{ij}^{\,2}, \cdots, h_{ij}^{\,k}\right\}\right\}}, f = 1, 2, 3, \cdots, k, \, j \in J_B \tag{21}
$$

**Table 2.**

| Criteria | DM | $A_1$ |
|---|---|---|
| $C_1$ | $D_1$ | VH |
| | $D_2$ | VH |
| | $D_3$ | HVH |
| | $D_4$ | HVH |

$$\bar{h}_{ij}^{\;f} = \frac{\min\limits_{1 \le i \le k}\left\{\min\left\{h_{ij}^{\;1}, h_{ij}^{\;2}, \cdots, h_{ij}^{\;k}\right\}\right\}}{h_{ij}^{\;f}}, f = 1, 2, 3, \cdots, k, j \in J_C \tag{22}$$

Where $J_B$ represents the benefit criteria set; $J_C$ represents the cost criteria set; $h_{ij}^{\;f}$ is the f-th evaluation value of the alternative $A_i$ under criterion $C_i$ in the hesitant fuzzy group decision matrix; and $k$ represents the number of decision experts.

**Step 7**: Construct PHF group decision matrix.

In view of the evaluation value of any scheme $A_i$ under any criterion $C_i$ in the standardized hesitant fuzzy group decision matrix, we assign each membership degree of the hesitant fuzzy element in the evaluation value to its corresponding probability according to its frequency of occurrence in the hesitant fuzzy element. For example, if the evaluation value of scheme $A_i$ is {0.32, 0.32, 0.56, 0.72} under the criterion $C_i$, then {0.32, 0.32, 0.56, 0.72} can be expressed as {0.32|0.5, 0.56|0.25, 0.72|0.25} by the PHFEs.

**Step 8**. Construct the ideal Alternative $A^*$.

Where $A^*$ can be defined as follows:

*For* $\forall f, g$ *and* $f \neq g, \exists s(h_{fj}(p_{fj})) \neq s(h_{gj}(p_{gj})),$

$$A^* = \left\{\left\langle x_j, h_{ij}(p_{ij})|\max_i\left(s(h_{ij}(p_{ij}))\right)\right\rangle | j = 1, 2, \cdots, m\right\} \tag{23}$$

*For* $\forall f, g$ *and* $f \neq g, \exists \max_i\left(s(h_{ij}(p_{ij}))\right) = s(h_{fj}(p_{fj})) = s(h_{gj}(p_{gj}))$

$$A^* = \left\{\left\langle x_j, h_{ij}(p_{ij})|\min_i \delta(h_{ij}(p_{ij}))\right\rangle | j = 1, 2, \cdots, m\right\} \tag{24}$$

Here, $s(h_{ij}(p_{ij}))$ and $\delta(h_{ij}(p_{ij}))$ are calculated using formulas (3) and (4), and and represent the score function value and deviation function value of the evaluation value of scheme $A_i$ under the j-th criterion in the PHF group decision matrix, respectively.

**Step 9**: Calculate the correlation coefficient between any alternative $A_i$ and $A^*$.

Using our newly proposed PHFSs weighted correlation coefficient formula (19), the correlation coefficients between any alternative $A_i$ and $A^*$ are calculated.

**Step 10**: Rank alternatives.

The alternatives are ranked according to the correlation coefficient calculated in step 9.

## 6. Case study

### 6.1. Problem description

Narcolepsy is a chronic sleep disorder of unknown cause that will be included in the list of rare diseases in China in 2023. However, owing to the lack of sufficient clinical data and clear treatment standards, it is difficult to measure the clinical value and economic evaluation of drugs for treating narcolepsy using the standards for ordinary drugs. This makes it difficult to use the traditional HTA for comprehensive clinical evaluation. In addition, compared to other ordinary drugs, during the comprehensive evaluation of orphan drugs for narcolepsy, it is necessary to fully consider the characteristics of rare patients, high production costs, difficulties in research and development, and limited alternatives, weakening the cost-effectiveness factor and paying more attention to multiple dimensions, such as safety, effectiveness, economy, and social value. However, due to the lack of clinical data and the small number of patients with narcolepsy, the evaluation of narcolepsy orphan drugs relies on subjective evaluations by

**Table 3. The detailed description of criteria.**

| Criteria | Description |
|---|---|
| Safety | whether patients will have adverse reactions and adverse events after taking the drugs. |
| Effectiveness | whether it can meet the requirements of preventing, treating, diagnosing patients' diseases and regulating physiological functions under the conditions of specified indications, usage and dosage. |
| Economy | the production cost and market price level of the drug |
| Social value | whether the drug can improve the quality of life of patients from the perspective of social ethics. |

medical experts. Assuming that four medical experts $D = \{D_1, D_2 \cdots D_4\}$ need to choose one orphan drug from five treatments for narcolepsy to be included in the medical insurance reimbursement drug list, the experts evaluated the five drugs $A_i(i = 1, 2 \cdots, 5)$ mainly based on four criteria: safety (C1), effectiveness (C2), economy (C3), and social value (C4). A detailed description of these criteria is provided in Table 3. The experts used linguistic terms for the evaluation. The weights $w_j(1, 2, \cdots, 4)$ of each criterion are unknown and independent of each other, and the weights $w_q(q = 1, 2, \cdots, 4)$ of the four experts are the same. The decision-making structure is illustrated in Fig 2.

## 6.2. Decision process

**Step 1**: Construct individual decision matrix of each medical expert.

The linguistic variables in Table 1 were used by the four medical experts to evaluate all the alternative narcolepsy orphan drugs, and then the individual decision matrix of each expert was provided. The details are listed in Tables 4–7:

**Step 2**: Each medical expert evaluates the importance of each criterion.

Each expert uses the language variables in Table 1 to evaluate the importance of each criterion and then gives the criterion evaluation matrix, as shown in Table 8.

**Step 3**: Obtain criteria weights.

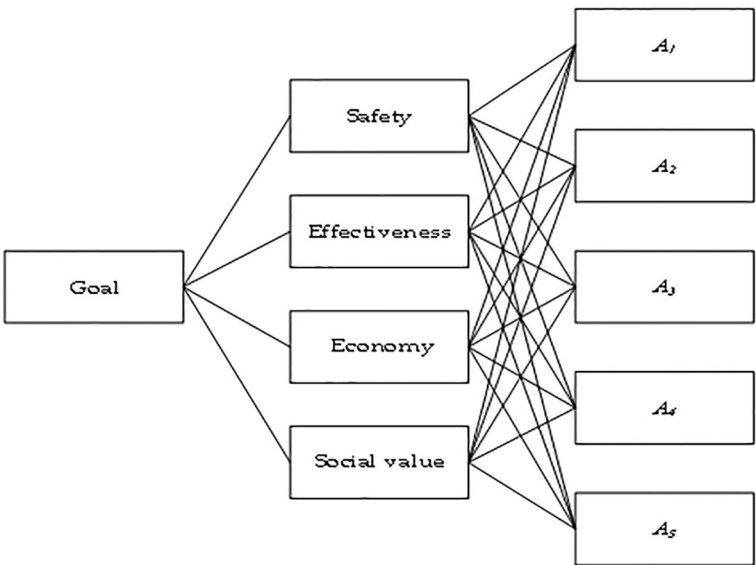

**Fig 2. Decision hierarchy structure.**

**Table 4. The individual decision matrix of D1.**

|        | $A_1$ | $A_2$ | $A_3$ | $A_4$ | $A_5$ |
|--------|-------|-------|-------|-------|-------|
| $C_1$  | VH    | L     | MH    | H     | H     |
| $C_2$  | M     | M     | H     | ML    | H     |
| $C_3$  | H     | H     | M     | ML    | MH    |
| $C_4$  | VH    | ML    | VL    | MH    | ML    |

**Table 5. The individual decision matrix of D2.**

|        | $A_1$ | $A_2$ | $A_3$ | $A_4$ | $A_5$ |
|--------|-------|-------|-------|-------|-------|
| $C_1$  | VH    | VLL   | H     | H     | H     |
| $C_2$  | ML    | MH    | H     | ML    | MH    |
| $C_3$  | H     | H     | M     | ML    | H     |
| $C_4$  | VH    | ML    | VL    | H     | ML    |

**Table 6. The individual decision matrix of D3.**

|        | $A_1$ | $A_2$ | $A_3$ | $A_4$ | $A_5$ |
|--------|-------|-------|-------|-------|-------|
| $C_1$  | HVH   | ML    | H     | MH    | H     |
| $C_2$  | ML    | H     | VH    | L     | MH    |
| $C_3$  | H     | H     | M     | L     | H     |
| $C_4$  | H     | M     | VLL   | MH    | M     |

**Table 7. The individual decision matrix of D4.**

|        | $A_1$ | $A_2$ | $A_3$ | $A_4$ | $A_5$ |
|--------|-------|-------|-------|-------|-------|
| $C_1$  | HVH   | ML    | H     | MH    | H     |
| $C_2$  | ML    | VH    | VH    | L     | H     |
| $C_3$  | MH    | H     | M     | L     | HVH   |
| $C_4$  | H     | MH    | VLL   | MH    | M     |

**Table 8. The criterion evaluation matrix.**

|        | $C_1$ | $C_2$ | $C_3$ | $C_4$ |
|--------|-------|-------|-------|-------|
| $D_1$  | M     | MH    | VH    | VL    |
| $D_2$  | M     | MH    | VH    | VL    |
| $D_3$  | L     | H     | HVH   | L     |
| $D_4$  | M     | HVH   | MH    | L     |

We referred to the practice of Chen and Xu [89] to obtain the criteria weights, and the specific algorithm is as follows:

We take the criterion $C_1$ as an example, where four experts assign the evaluation values of the weight of the criterion $C_1$ as M,M,L, and M. Then, these evaluation values are converted

**Table 9. The criteria weights matrix.**

| $C_1$ | $C_2$ | $C_3$ | $C_4$ |
|---|---|---|---|
| 0.21 | 0.31 | 0.39 | 0.09 |

into 5,5,3,5 according to Saaty's 1–9 scale. Next, according to the weights of the four experts, the integrated weight of criterion $C_1$ is $w_1 = 5{\times}0.25+5{\times}0.25+3{\times}0.25+5{\times}0.25 = 4.5$. Using this method, we calculated the integrated weights of all criteria and standardized the integrated weights of all criteria. Subsequently, we obtained the final criteria weights according to the above steps. The results are presented in Table 9.

**Step 4**: Construct individual weighted numerical decision matrix.

According to the scale in Table 1, the individual decision matrix was transformed into a personal weighted numerical decision matrix according to the expert weight, and the results are shown in Tables 10–13:

**Table 10. The individual weighted numerical decision matrix of D1.**

| | $A_1$ | $A_2$ | $A_3$ | $A_4$ | $A_5$ |
|---|---|---|---|---|---|
| $C_1$ | 2.25 | 0.75 | 1.5 | 1.75 | 1.75 |
| $C_2$ | 1.25 | 1.25 | 1.75 | 1 | 1.75 |
| $C_3$ | 1.75 | 1.75 | 1.25 | 1 | 1.5 |
| $C_4$ | 2.25 | 1 | 0.25 | 1.5 | 1 |

**Table 11. The individual weighted numerical decision matrix of D2.**

| | $A_1$ | $A_2$ | $A_3$ | $A_4$ | $A_5$ |
|---|---|---|---|---|---|
| $C_1$ | 2.25 | 0.5 | 1.75 | 1.75 | 1.75 |
| $C_2$ | 1 | 1.5 | 1.75 | 1 | 1.5 |
| $C_3$ | 1.75 | 1.75 | 1.25 | 1 | 1.75 |
| $C_4$ | 2.25 | 1 | 0.25 | 1.75 | 1 |

**Table 12. The individual weighted numerical decision matrix of D3.**

| | $A_1$ | $A_2$ | $A_3$ | $A_4$ | $A_5$ |
|---|---|---|---|---|---|
| $C_1$ | 2 | 1 | 1.75 | 1.5 | 1.75 |
| $C_2$ | 1 | 1.75 | 2.25 | 0.75 | 1.5 |
| $C_3$ | 1.75 | 1.75 | 1.25 | 0.75 | 1.75 |
| $C_4$ | 1.75 | 1.25 | 0.5 | 1.5 | 1.25 |

**Table 13. The individual weighted numerical decision matrix of D4.**

| | $A_1$ | $A_2$ | $A_3$ | $A_4$ | $A_5$ |
|---|---|---|---|---|---|
| $C_1$ | 2 | 1 | 1.75 | 1.5 | 1.75 |
| $C_2$ | 1 | 2.25 | 2.25 | 0.75 | 1.75 |
| $C_3$ | 1.5 | 1.75 | 1.25 | 0.75 | 2 |
| $C_4$ | 1.75 | 1.5 | 0.5 | 1.5 | 1.25 |

**Table 14. The hesitant fuzzy group decision matrix.**

|  | $C_1$ | $C_2$ | $C_3$ | $C_4$ |
|---|---|---|---|---|
| $A_1$ | 2,2,2.25,2.25 | 1,1,1,1.25 | 1.5,1.75,1.75,1.75 | 1.75,1.75,2.25,2.25 |
| $A_2$ | 0.5,0.75,1,1 | 1.25,1.5,1.75,2.25 | 1.75,1.75,1.75,1.75 | 1,1,1.25,1.5 |
| $A_3$ | 1.5,1.75,1.75,1.75 | 1.75,1.75,2.25,2.25 | 1.25,1.25,1.25,1.25 | 0.25,0.25,0.5,0.5 |
| $A_4$ | 1.5,1.5,1.75,1.75 | 0.75,0.75,1,1 | 0.75,0.75,1,1 | 1.5,1.5,1.5,1.75 |
| $A_5$ | 1.75,1.75,1.75,1.75 | 1.5,1.5,1.75,1.75 | 1.5,1.75,1.75,2 | 1,1,1.25,1.25 |

**Step 5**: Construct hesitant fuzzy group decision matrix.

The individual weighted numerical decision matrix of each expert in Step 4 is integrated into the hesitant fuzzy group decision matrix, and the specific results are shown in Table 14.

**Step 6**: Standardize hesitant fuzzy group decision matrix.

Referring to the practice of Chen and Xu [89], we standardized the hesitating fuzzy group decision matrix by using formulas (21) and (22), where $C_2$ is the cost criterion and the other criteria are the benefit criteria. The results are presented in Table 15.

**Step 7**: Construct PHF group decision matrix.

The standardized hesitant fuzzy group decision matrix in Step 6 is transformed into the PHF group decision matrix, and the specific results are shown in Table 16.

**Step 8**: Construct the ideal alternative $A^*$.

Using formulas (23) and (24) to construct an ideal alternative, $A^*$, the specific results are shown in Table 17.

**Step 9**: Calculate the correlation coefficient between any alternative $A_i$ and $A^*$.

**Table 15. The normalized hesitant fuzzy group decision matrix.**

|  | $C_1$ | $C_2$ | $C_3$ | $C_4$ |
|---|---|---|---|---|
| $A_1$ | 0.89,0.89,1,1 | 0.75,0.75,0.75,0.60 | 0.75,0.88,0.88,0.88 | 0.78,0.78,1,1 |
| $A_2$ | 0.22,0.33,0.44,0.44 | 0.60,0.50,0.43,0.33 | 0.88,0.88,0.88,0.88 | 0.44,0.44,0.56,0.67 |
| $A_3$ | 0.67,0.78,0.78,0.78 | 0.43,0.43,0.33,0.33 | 0.63,0.63,0.63,0.63 | 0.11,0.11,0.22,0.22 |
| $A_4$ | 0.67,0.67,0.78,0.78 | 1.00,1.00,0.75,0.75 | 0.38,0.38,0.50,0.50 | 0.67,0.67,0.67,0.78 |
| $A_5$ | 0.78,0.78,0.78,0.78 | 0.50,0.50,0.43,0.43 | 0.75,0.88,0.88,1.00 | 0.44,0.44,0.56,0.56 |

**Table 16. The PHF group decision matrix.**

|  | $C_1$ | $C_2$ | $C_3$ | $C_4$ |
|---|---|---|---|---|
| $A_1$ | 0.89\|0.5,1\|0.5 | 0.6\|0.25,0.75\|0.75 | 0.75\|0.25,0.88\|0.75 | 0.78\|0.5, 1\|0.5 |
| $A_2$ | 0.22\|0.25, 0.33\|0.25,0.44\|0.5 | 0.33\|0.25,0.43\|0.25, 0.5\|0.25,0.6\|0.25 | 0.88\|1 | 0.56\|0.25, 0.67\|0.25,0.44\|0.5 |
| $A_3$ | 0.67\|0.25,0.78\|0.75 | 0.33\|0.5,0.43\|0.5 | 0.63\|1 | 0.11\|0.5,0.22\|0.5 |
| $A_4$ | 0.67\|0.5,0.78\|0.5 | 0.75\|0.5,1\|0.5 | 0.38\|0.5,0.5\|0.5 | 0.78\|0.25,0.67\|0.75 |
| $A_5$ | 0.78\|1 | 0.43\|0.5,0.5\|0.5 | 0.75\|0.25, 1\|0.25,0.88\|0.5 | 0.44\|0.5,0.56\|0.5 |

**Table 17. The ideal alternative $A^*$.**

|  | $C_1$ | $C_2$ | $C_3$ | $C_4$ |
|---|---|---|---|---|
| $A^*$ | 0.89\|0.5,1\|0.5 | 0.75\|0.5,1\|0.5 | 0.88\|1 | 0.78\|0.5,1\|0.5 |

Table 18. The calculation result of correlation coefficient.

| $C(A_1, A^*)$ | $C(A_2, A^*)$ | $C(A_3, A^*)$ | $C(A_4, A^*)$ | $C(A_5, A^*)$ |
|---|---|---|---|---|
| 0.7708 | 0.8932 | 0.9589 | 0.6794 | 0.4699 |

Formula (19) is used to calculate the correlation coefficient between each alternative, $A_i$ and $A^*$, and the specific results are shown in Table 18.

**Step 10**: Rank alternatives.

$$A_3 \succ A_2 \succ A_1 \succ A_4 \succ A_5.$$

We can see that since the correlation coefficient between $A_3$ and the ideal alternative $A^*$ is the largest, $A_3$ is the best alternative, and will be selected as the drug to enter the reimbursement list of medical insurance.

# 7. Discussion

In this section, to observe the impact of different criteria weights on the decision results of the proposed method, we first conduct a sensitivity analysis of the criteria weights. Second, to illustrate the feasibility and efficacy of our novel correlation coefficient and the MCGDM method proposed based on the correlation coefficient, we first conduct a test on the existence of the rank reversal phenomenon. In recent years, the rank reversal phenomenon in decision analysis has become a research focus of some scholars. The rank reversal phenomenon is generally caused by the following factors: 1. adding a worse scheme; 2. adding a better scheme; and 3. adding a scheme that performs similarly to existing schemes, and 4. deleting a scheme. To overcome the rank reversal phenomenon, some scholars have proposed decision methods, such as COMET [72], ESP-COMET [73], SPOTIS [74], and SIMUS [75]. Therefore, it was necessary to conduct a test on the rank reversal phenomenon to demonstrate the reliability of the proposed method. Finally, we compared and analyzed our method with other existing correlation coefficients and the corresponding MCDM methods.

## 7.1 Sensitivity analysis of criteria weights

The proposed decision method was obtained by obtaining the objective weights. To observe the influence of weights on decision results, we set different criteria weights to observe changes in the decision results. The results are presented in Table 19.

From Table 19 and Fig 3, we can see that the results obtained by setting different criteria weight vectors are consistent with our original decision results. The optimal alternative is $A_3$ and the worst alternative is $A_5$, indicating that our proposed method is essentially unaffected by weights.

Table 19. Ranking results for different criteria weights.

| $w$ | $C(A_1, A^*)$ | $C(A_2, A^*)$ | $C(A_3, A^*)$ | $C(A_4, A^*)$ | $C(A_5, A^*)$ | Ranking |
|---|---|---|---|---|---|---|
| $w_1 = (0.1, 0.2, 0.3, 0.4)$ | 0.8031 | 0.8721 | 0.9218 | 0.7149 | 0.5703 | $A_3 \succ A_2 \succ A_1 \succ A_4 \succ A_5$ |
| $w_2 = (0.2, 0.1, 0.3, 0.4)$ | 0.8092 | 0.8738 | 0.9153 | 0.7199 | 0.5089 | $A_3 \succ A_2 \succ A_1 \succ A_4 \succ A_5$ |
| $w_3 = (0.3, 0.2, 0.4, 0.1)$ | 0.7740 | 0.8981 | 0.9555 | 0.6805 | 0.4402 | $A_3 \succ A_2 \succ A_1 \succ A_4 \succ A_5$ |
| $w_4 = (0.4, 0.3, 0.2, 0.1)$ | 0.8432 | 0.8080 | 0.9240 | 0.7912 | 0.4805 | $A_3 \succ A_1 \succ A_2 \succ A_4 \succ A_5$ |

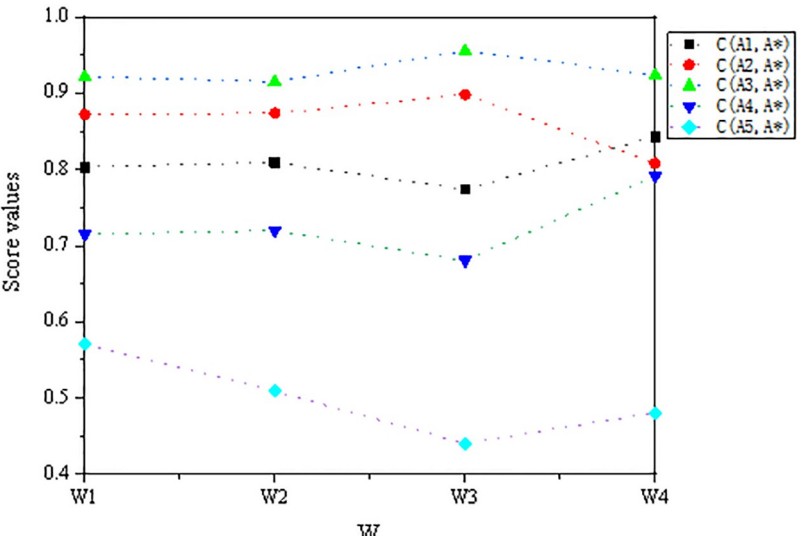

**Fig 3. Ranking results for different criteria weights.**

**Table 20. Decision matrix after adding new alternative $A_6$.**

|  | $C_1$ | $C_2$ | $C_3$ | $C_4$ |
|---|---|---|---|---|
| $A_1$ | 0.89\|0.5,1\|0.5 | 0.6\|0.25,0.75\|0.75 | 0.75\|0.25,0.88\|0.75 | 0.78\|0.5, 1\|0.5 |
| $A_2$ | 0.22\|0.25, 0.33\|0.25,0.44\|0.5 | 0.33\|0.25,0.43\|0.25, 0.5\|0.25,0.6\|0.25 | 0.88\|1 | 0.56\|0.25, 0.67\|0.25,0.44\|0.5 |
| $A_3$ | 0.67\|0.25,0.78\|0.75 | 0.33\|0.5,0.43\|0.5 | 0.63\|1 | 0.11\|0.5,0.22\|0.5 |
| $A_4$ | 0.67\|0.5,0.78\|0.5 | 0.75\|0.5,1\|0.5 | 0.38\|0.5,0.5\|0.5 | 0.78\|0.25,0.67\|0.75 |
| $A_5$ | 0.78\|1 | 0.43\|0.5,0.5\|0.5 | 0.75\|0.25, 1\|0.25,0.88\|0.5 | 0.44\|0.5,0.56\|0.5 |
| $A_6$ | 0.63\|0.25,0.76\|0.75 | 0.3\|0.5,0.4\|0.5 | 0.6\|1 | 0.1\|0.5,0.2\|0.5 |

## 7.2 Test on the phenomenon of rank reversal

To verify whether our proposed method exhibits rank reversal, we added an alternative that is close to the original optimal alternative $A_3$ and another alternative that is close to the worst scheme $A_5$. We name these two alternatives $A_6$ and $A_6^+$, respectively, where $A_6$ is set as a slightly inferior alternative to $A_3$, that is, $A_6 \succ A_6$, and $A_6^+$ is set as a slightly inferior alternative to $A_5$, that is, $A_5 \succ A_6^+$. Then, we observe whether adding $A_6$ and $A_6^+$ separately affects the order between the original alternatives, that is, keeping the order $A_3 \succ A_2 \succ A_1 \succ A_4 \succ A_5$ unchanged. The following probabilistic hesitant fuzzy decision matrices after adding new alternatives $A_6$ and $A_6^+$, respectively, are detailed in Tables 20 and 21:

According to the above decision matrix, and then using our proposed method, the following decision results are obtained, as shown in Table 22:

After adding $A_6$ and $A_6^+$, the ranking among the original alternatives, that is, $A_3 \succ A_2 \succ A_1 \succ A_4 \succ A_5$, does not change, which indicates that there is no rank reversal in our proposed method, which indicates that our proposed method has applicability and reliability in MCDM.

**Table 21. Decision matrix after adding new alternative $A_6^+$.**

|  | $C_1$ | $C_2$ | $C_3$ | $C_4$ |
|---|---|---|---|---|
| $A_1$ | 0.89\|0.5,1\|0.5 | 0.6\|0.25,0.75\|0.75 | 0.75\|0.25,0.88\|0.75 | 0.78\|0.5, 1\|0.5 |
| $A_2$ | 0.22\|0.25, 0.33\|0.25,0.44\|0.5 | 0.33\|0.25,0.43\|0.25, 0.5\|0.25,0.6\|0.25 | 0.88\|1 | 0.56\|0.25, 0.67\|0.25,0.44\|0.5 |
| $A_3$ | 0.67\|0.25,0.78\|0.75 | 0.33\|0.5,0.43\|0.5 | 0.63\|1 | 0.11\|0.5,0.22\|0.5 |
| $A_4$ | 0.67\|0.5,0.78\|0.5 | 0.75\|0.5,1\|0.5 | 0.38\|0.5,0.5\|0.5 | 0.78\|0.25,0.67\|0.75 |
| $A_5$ | 0.78\|1 | 0.43\|0.5,0.5\|0.5 | 0.75\|0.25, 1\|0.25,0.88\|0.5 | 0.44\|0.5,0.56\|0.5 |
| $A_6^+$ | 0.75\|1 | 0.4\|0.5,0.45\|0.5 | 0.7\|0.25, 0.8\|0.25,0.85\|0.5 | 0.4\|0.5,0.5\|0.5 |

**Table 22. Decision results after adding $A_6$ and $A_6^+$.**

| $A_6$ | $A_3 \succ A_6 \succ A_2 \succ A_1 \succ A_4 \succ A_5$ |
|---|---|
| $A_6^+$ | $A_3 \succ A_2 \succ A_1 \succ A_4 \succ A_5 \succ A_6^+$ |

## 7.3 Comparative analysis with other correlation coefficients and MCDM methods

To illustrate the feasibility and efficacy of our novel correlation coefficient and the MCGDM method proposed on the basis of the correlation coefficient, we compared the proposed correlation coefficient with the three correlation coefficients proposed in the existing literature and their application effects in the corresponding MCDM. First, we compare the proposed correlation coefficient with the mean correlation coefficient proposed by Wang and Li [53]. Second, we compare the proposed correlation coefficient with the mixed correlation coefficient proposed by Liu and Guan [55]. Finally, to illustrate the advancement of PHFSs compared with HFSs,we compare the proposed correlation coefficients with those proposed by Chen et al. [45] without considering DMs' preferences.

First, we compare the proposed correlation coefficient with the mean correlation coefficient proposed by Wang and Li [53]. The specific process is that we use the same calculation example and criteria weights (0.19,0.21,0.19,0.20,0.21) in the literature [53] and use the correlation coefficient and decision method proposed by us to make decisions on this calculation example. The decision matrix is shown in Table 23.

The specific decision-making process is as follows:

**Step 1**: Construct the ideal alternative $A^*$.

According to Eqs (23) and (24), we constructed the ideal alternative $A^*$, and the specific results are shown in Table 24.

**Table 23. The decision matrix.**

|  | $C_1$ | $C_2$ | $C_3$ | $C_4$ | $C_5$ |
|---|---|---|---|---|---|
| $A_1$ | 0.7\|0.25, 0.5\|0.75 | 0.8\|0.1, 0.6\| 0.25, 0.7\|0.65 | 0.3\|0.15, 0.6\|0.25, 0.4\| 0.6 | 0.3\|0.125, 0.4\|0.125, 0.6\|0.25, 0.7\|0.5 | 0.3\|0.25, 0.4\|0.25, 0.5\| 0.25, 0.8\|0.25 |
| $A_2$ | 0.2\|0.1, 0.4\|0.25, 0.3\| 0.65 | 0.4\|0.5, 0.5\|0.5 | 0.6\|0.125, 0.4\|0.25, 0.5\| 0.375, 0.8\|0.25 | 0.4\|0.25, 0.6\|0.375, 0.7\|0.375 | 0.4\|0.25, 0.6\|0.75 |
| $A_3$ | 0.4\|0.25, 0.7\|0.3, 0.6\| 0.45 | 0.4\|0.25, 0.6\| 0.25, 0.5\|0.5 | 0.2\|0.25, 0.3\|0.25, 0.5\| 0.25, 0.6\|0.25 | 0.1\|0.25, 0.5\|0.25, 0.3\| 0.5 | 0.6\|0.125, 0.4\|0.5, 0.7\| 0.375 |
| $A_4$ | 0.2\|0.25, 0.3\|0.25, 0.4\|0.25, 0.6\|0.25 | 0.4\|0.25, 0.3\| 0.5, 0.6\|0.25 | 0.3\|0.25, 0.4\|0.25, 0.5\| 0.5 | 0.4\|0.175, 0.6\|0.25, 0.5\|0.575 | 0.2\|0.25, 0.4\|0.125, 0.5\| 0.375, 0.7\|0.25 |

**Table 24. The ideal alternative $A^*$.**

|       | $C_1$ | $C_2$ | $C_3$ | $C_4$ | $C_5$ |
|-------|-------|-------|-------|-------|-------|
| $A^*$ | 0.4\|0.25, 0.7\|0.3, 0.6\| 0.45 | 0.8\|0.1, 0.6\|0.25, 0.7\| 0.65 | 0.3\|0.15, 0.6\|0.25, 0.4\|0.6 | 0.4\|0.25, 0.6\|0.375, 0.7\| 0.375 | 0.4\|0.25, 0.6\| 0.75 |

**Table 25. The calculation result of correlation coefficient.**

| $C(A_1, A^*)$ | $C(A_2, A^*)$ | $C(A_3, A^*)$ | $C(A_4, A^*)$ |
|---------------|---------------|---------------|---------------|
| 0.7755 | 0.9092 | 0.8626 | 0.7706 |

**Step 2**: Calculate the correlation coefficient between any alternative $A_i$ and $A^*$.
The specific results are shown in Table 25:
**Step 3**: Rank alternatives.

$$A_2 \succ A_3 \succ A_1 \succ A_4$$

The final ranking result obtained by us was $A_2 \succ A_3 \succ A_1 \succ A_4$. As shown in Fig 4, this result is consistent with the results of the method proposed by Wang and Li [53], indicating the feasibility of the proposed correlation coefficient and decision-making method. However, compared to the mean correlation coefficient proposed by Wang and Li [53], the proposed correlation coefficient can compensate for the deficiency of the mean correlation coefficient; that is, if the mean value of each PHFE between multiple PHFSs is equal, it can be concluded that the correlation coefficient between multiple PHFSs is equal to 1.

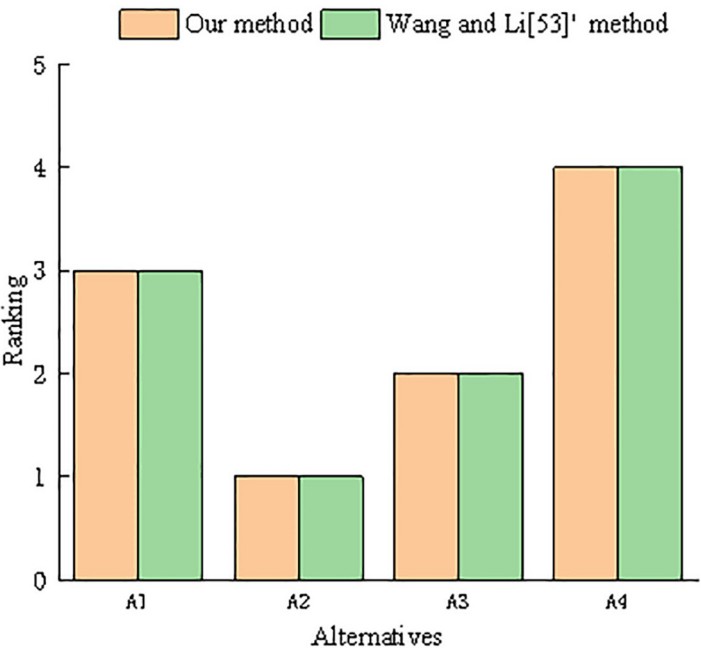

**Fig 4. Comparison of results between our method and Wang and Li [53]'s method.**

**Table 26. The decision matrix.**

|        | $C_1$ | $C_2$ | $C_3$ |
|--------|-------|-------|-------|
| $A_1$ | 0.76\|0.1,0.55\|0.15, 0.65\|0.25,0.8\|0.5 | 0.2\|0.05,0.4\|0.125, 0.3\|0.325,0.65\|0.25, 0.75\|0.25 | 0.94\|0.1,0.8\|0.15, 0.75\|0.25,0.55\|0.5 |
| $A_2$ | 0.4\|0.25,0.58\|0.25, 0.69\|0.25,0.95\|0.25 | 0.6\|0.075,0.8\|0.075, 0.35\|0.25,0.65\|0.25, 0.7\|0.35 | 0.25\|0.25,0.45\|0.375, 0.65\|0.375 |
| $A_3$ | 0.3\|0.1,0.68\|0.25, 0.5\|0.35,0.6\|0.3 | 0.55\|0.125,0.66\|0.125, 0.45\|0.25,0.56\|0.25, 0.85\|0.25 | 0.45\|0.25,0.55\|0.25, 0.68\|0.25,0.75\|0.25 |
| $A_4$ | 0.15\|0.1,0.37\|0.15, 0.4\|0.25,0.6\|0.25, | 0.62\|0.1,0.55\|0.25, 0.66\|0.25,0.48\|0.4 | 0.5\|0.125,0.7\|0.125, 0.38\|0.25,0.75\|0.25, |
|        | 0.73\|0.25 |  | 0.85\|0.25 |

Next, we compare the proposed correlation coefficient with the mixed correlation coefficient proposed by Liu and Guan [55]. The specific process is as follows: We adopt the same calculation examples and criteria weights as in the literature [55], namely (0.39,0.26,0.35), and then use the correlation coefficient and decision method proposed by us to make decisions. The decision matrices are listed in Table 26.

The specific decision-making process is as follows:

**Step 1**: Construct the ideal alternative $A^*$.

According to Eqs (23) and (24), we construct ideal scheme $A^*$, as shown in Table 27:

**Step 2**: Calculate the correlation coefficient between any alternative $A_i$ and $A^*$.

The specific results are shown in Table 28:

**Step 3**: Rank alternatives.

$$A_1 \succ A_3 \succ A_2 \succ A_4.$$

As shown in Fig 5, by comparing with the sorting results $A_1 \succ A_2 \succ A_3 \succ A_4$ in the literature [55], we find that the optimal scheme $A_1$ and the worst scheme $A_4$ are the same as the sorting results in the literature [55]. However, the sorting results between $A_2$ and $A_3$ are different because the mixed correlation coefficient proposed by Liu and Guan [55] needs to subjectively set the weights for the mean, variance, and length rate correlation coefficients, which will lead to the subjectivity of the decision-making results to a certain extent. However, the proposed correlation coefficient completely depends on objective evaluation information and does not need to set the weights subjectively. Therefore, the proposed correlation coefficient has the advantage of making the decision results unique and objective.

**Table 27. The ideal alternative $A^*$.**

|        | $C_1$ | $C_2$ | $C_3$ |
|--------|-------|-------|-------|
| $A^*$ | 0.76\|0.1,0.55\|0.15, 0.65\|0.25,0.8\|0.5 | 0.62\|0.1,0.55\|0.25, 0.66\|0.25,0.48\|0.4 | 0.94\|0.1,0.8\|0.15, 0.75\|0.25,0.55\|0.5 |

**Table 28. The calculation result of correlation coefficient.**

| $C(A_1, A^*)$ | $C(A_2, A^*)$ | $C(A_3, A^*)$ | $C(A_4, A^*)$ |
|---------------|---------------|---------------|---------------|
| 0.9564 | 0.8517 | 0.8667 | 0.5799 |

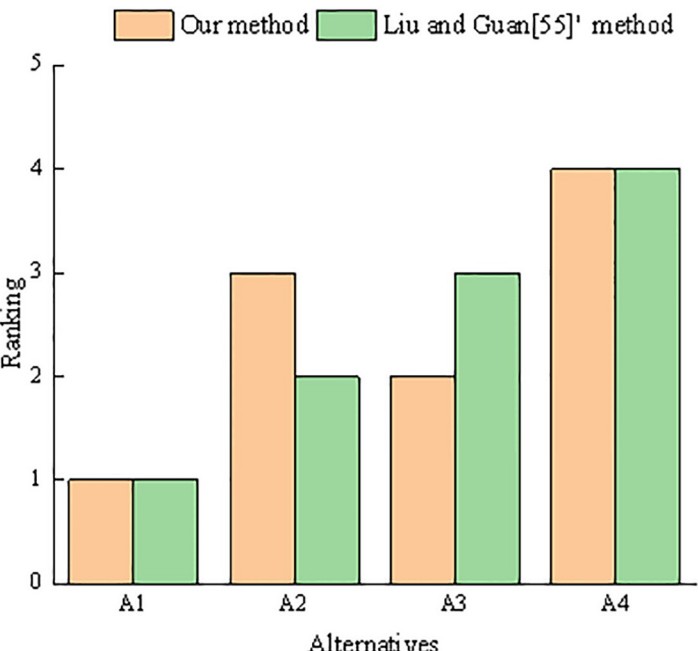

**Fig 5. Comparison of results between our method and Liu and Guan [55]'s method.**

Next, we compare the proposed correlation coefficient with that proposed by Chen et al. [45] without considering the preference of DMs. The specific process is as follows: First, we remove the preference information of DMs, that is, the probability information, from the PHF group decision matrix obtained in the fifth part and then use the HFS correlation coefficient proposed by Chen et al. [45]. In addition, the same criteria weights (0.21,0.31,0.39,0.09) as in the calculation examples in the fifth part of this study were adopted to make decisions by using steps similar to the decision method proposed by us. The specific decision matrices are presented in Table 29.

The specific decision-making process is as follows:

**Step 1**: Construct the ideal alternative $A^*$.

According to the scoring function and deviation function proposed in the literature [19], and using the same idea as formulas (23) and (24), we constructed an ideal alternative $A^*$ in a hesitant fuzzy environment, in which the details are as shown in Table 30.

**Step 2**: Calculate the correlation coefficient between any alternative $A_i$ and $A^*$.

The specific results are shown in Table 31:

**Table 29. The decision matrix.**

|  | $C_1$ | $C_2$ | $C_3$ | $C_4$ |
|---|---|---|---|---|
| $A_1$ | 0.89,1 | 0.60,0.75 | 0.75,0.88 | 0.78,1 |
| $A_2$ | 0.22,0.33,0.44 | 0.33,0.43,0.50,0.60 | 0.88 | 0.44,0.56,0.67 |
| $A_3$ | 0.67,0.78 | 0.33,0.43 | 0.63 | 0.11,0.22 |
| $A_4$ | 0.67,0.78 | 0.75,1.00 | 0.38,0.50 | 0.67,0.78 |
| $A_5$ | 0.78 | 0.43,0.50 | 0.75,0.88,1.00 | 0.44,0.56 |

**Table 30. The ideal alternative $A^*$.**

|  | $C_1$ | $C_2$ | $C_3$ | $C_4$ |
|---|---|---|---|---|
| $A^*$ | 0.89,1 | 0.75,1 | 0.75,0.88 | 0.78,1 |

**Table 31. The calculation result of correlation coefficient.**

| $C(A_1, A^*)$ | $C(A_2, A^*)$ | $C(A_3, A^*)$ | $C(A_4, A^*)$ | $C(A_5, A^*)$ |
|---|---|---|---|---|
| 0.9943 | 0.8941 | 0.9108 | 0.9804 | 0.9515 |

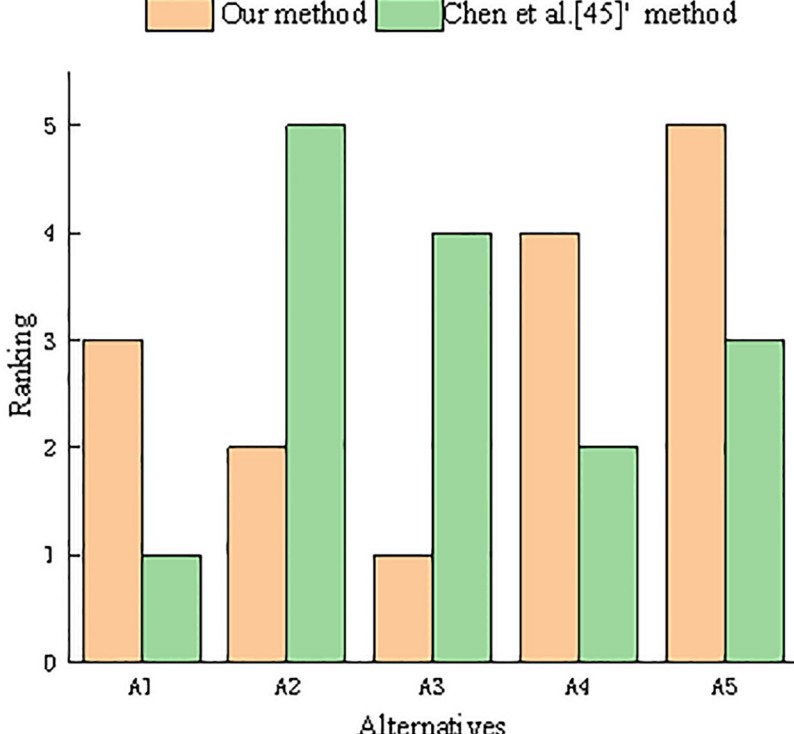

**Fig 6. Comparison of results between our method and Chen et al. [45]'s method.**

**Step 3**: Rank alternatives.

$$A_1 \succ A_4 \succ A_5 \succ A_3 \succ A_2.$$

As shown in Fig 6, it is clear that the decision result is different from the result of the $A_3 \succ A_2 \succ A_1 \succ A_4 \succ A_5$ of our proposed method. In a hesitant fuzzy environment, the optimal solution is $A_3$, whereas in a PHF environment, it is $A_1$. The difference in the result is mainly caused by the DM's preference information, which is also known as probabilistic information, which makes the decision information fully expressed and more consistent with the DM's thinking habit process, thereby enhancing the accuracy of the decision outcomes. Therefore, from this point of view, the newly proposed correlation coefficient in a PHF environment and

the proposed decision-making method based on the correlation coefficient appear to be more effective and credible than the correlation coefficient and decision-making method in a hesitant fuzzy environment.

## 8. Conclusion

In this study, we introduce PHFSs into the clinical comprehensive evaluation of orphan drugs to address the shortcomings of the existing MCDM method, which does not account for the uncertainty, fuzziness, and hesitation of experts in the decision-making process. Then, under a probabilistic hesitant fuzzy environment, we introduce information energy into the PHFSs to address the deficiencies of the existing PHFS correlation coefficients, and propose some new correlation coefficients for PHFSs, as well as the weighted form of correlation coefficients, and prove their properties. Subsequently, considering that medical experts are accustomed to using linguistic variables when evaluating different criteria for orphan drugs, we propose a method to transform the evaluation information of language variables into PHF evaluation information. Then, based on this method, we obtained the PHF group decision-making matrix and the weights of each evaluation criterion. Based on the above research, we extend the correlation coefficient proposed above to MCGDM and propose an MCGDM method based on the correlation coefficient under a PHF environment and unknown weights. To demonstrate the practicability of our proposed approach, we applied the newly proposed MCGDM method for the comprehensive clinical evaluation of orphan drugs. Finally, to verify the reliability, feasibility, and effectiveness of the new correlation coefficient and MCGDM method proposed in this study, we conducted a sensitivity analysis on the criteria weights of the proposed method. Then, to verify that the proposed method does not exhibit the phenomenon of rank reversal, we add some new schemes that are close to the optimal scheme and the worst scheme to investigate whether rank reversal will occur. Finally, we compared the newly proposed correlation coefficients with three other existing correlation coefficients and their corresponding MCDM methods. The results demonstrate that the proposed correlation coefficient is superior to the previous correlation coefficients. Compared to the correlation coefficient of HFSs, our suggested correlation coefficient of PHFSs compensates for the absence of preference information in DMs through the addition of probability information. However, when compared to the correlation coefficient of existing PHFSs, it adapts to more PHFS situations, and the calculation results are unaffected by the subjective weight setting. Based on these benefits, the proposed MCGDM approach is practical and efficient. However, the proposed method had several limitations. For example, during the decision-making process, some experts will express neutrality or opposition in the evaluation; in this case, PHFSs cannot represent such experts' expression information. As a result, our decision-making method in the PHF environment has some limitations.

In the future research, we will further explore another fuzzy sets that can contain the neutral and opposition information of DMs, such as probabilistic picture hesitant fuzzy sets (P-PHFSs), etc., and then explore new correlation coefficients under P-PHFSs, construct new fuzzy MCGDM methods, and broaden the application scope of correlation coefficients, such as fuzzy clustering algorithms, pattern recognition and classification, and we will further study it in the future.

## Supporting information

**S1 Table. The individual decision matrix of D1.**
(DOC)

**S2 Table. The individual decision matrix of D2.**
(DOC)

**S3 Table. The individual decision matrix of D3.**
(DOC)

**S4 Table. The individual decision matrix of D4.**
(DOC)

**S5 Table. The PHF group decision matrix.**
(DOC)

**S6 Table. Decision matrix after adding new alternative $A_6$.**
(DOC)

**S7 Table. Decision matrix after adding new alternative $A_6{}^+$.**
(DOC)

**S8 Table. The decision matrix.**
(DOC)

**S9 Table. The decision matrix.**
(DOC)

**S10 Table. The decision matrix.**
(DOC)

## Acknowledgments

The authors wish to thank anonymous reviewers for their valuable suggestions.

## Author Contributions

**Conceptualization:** Yubo Hu, Zhiqiang Pang.

**Formal analysis:** Yubo Hu, Zhiqiang Pang.

**Investigation:** Yubo Hu, Zhiqiang Pang.

**Methodology:** Yubo Hu.

**Validation:** Yubo Hu.

**Visualization:** Yubo Hu.

**Writing – original draft:** Yubo Hu, Zhiqiang Pang.

**Writing – review & editing:** Yubo Hu.

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
