## [Decision Letter · Decision Letter 0]

20 Dec 2023

PONE-D-23-33158A novel MCGDM technique based on correlation coefficients under probabilistic hesitant fuzzy environment and its application in clinical comprehensive evaluation of orphan drugsPLOS ONE

Dear Dr. Hu,

Thank you for submitting your manuscript to PLOS ONE. After careful consideration, we feel that it has merit but does not fully meet PLOS ONE’s publication criteria as it currently stands. Therefore, we invite you to submit a revised version of the manuscript that addresses the points raised during the review process.

We look forward to receiving your revised manuscript.

Kind regards,

Fausto Cavallaro, PhD

Academic Editor

PLOS ONE

Journal Requirements:

"The authors declare no conflict of interest."

Additional Editor Comments:

The reviewers were not positive with your paper. Once the reviewers comments are carefully addressed the paper can be assessed again.

Reviewers' comments:

Reviewer's Responses to Questions

**Comments to the Author**

1. Is the manuscript technically sound, and do the data support the conclusions?

Reviewer #1: Yes

Reviewer #2: No

Reviewer #3: Yes

2. Has the statistical analysis been performed appropriately and rigorously? 

Reviewer #1: N/A

Reviewer #2: N/A

Reviewer #3: Yes

3. Have the authors made all data underlying the findings in their manuscript fully available?

Reviewer #1: Yes

Reviewer #2: Yes

Reviewer #3: No

4. Is the manuscript presented in an intelligible fashion and written in standard English?

Reviewer #1: Yes

Reviewer #2: No

Reviewer #3: Yes

5. Review Comments to the Author

Reviewer #1: The scientific article titled "A novel MCGDM technique based on correlation coefficients under probabilistic hesitant fuzzy environment and its application in clinical comprehensive evaluation of orphan drugs" tackles the intricate task of decision-making within the nuanced context of probabilistic hesitant fuzzy sets (PHFSs). The manuscript strides into the realm of improving decision-making methodologies by proposing new correlation coefficients designed to address the limitations of existing ones for PHFSs, which is indeed a complex and forward-looking objective. The manuscript presents a promising and relevant study within the field of clinical evaluation, particularly the assessment of orphan drugs. It builds on the premise that PHFSs can overcome the problem of preference information loss—an issue not negligible in decision-making frameworks. The authors' initiative to advance the MCGDM method by incorporating a mechanism to convert linguistic variables into probabilistic hesitant fuzzy information suggests a nuanced understanding of the actual decision-making processes and showcases the manuscript's innovative aspect. Notwithstanding the paper's potential, several areas need to be refined for it to reach its full scholarly impact. The language requires meticulous proofreading by a native speaker to meet the academic standards expected for publication. Moreover, the article lacks a distinctive presentation of novelty; there is a noted absence of a compelling argument that clearly defines what sets this research apart from existing studies, a delineation of its unique contributions, and a discussion of its limitations. The manuscript should offer a more expansive comparison to contemporary methods such as "PT-TOPSIS methods for multi-attribute group decision making under single-valued neutrosophic sets" or "EDAS method for multiple attribute group decision making under spherical fuzzy environment," highlighting both similarities and differences. In addition, a broader literature review and a more comprehensive background on MCDA would fortify the study's contextual relevance and scholarly depth. The inclusion of a succinct comparison with methodologies such as SPOTIS, ESP-COMET, SIMUS, TOPSIS-DARIA, RANCOM, and others, is essential to demonstrate the robustness and relevance of the proposed MCGDM technique. Such an analysis would allow the authors to position their method within the current landscape of MCDA tools, revealing its potential advantages, limitations, and differentiating factors. Moreover, this comparison is pivotal to emphasize the novelty and original contribution of the work, as it showcases how the proposed technique performs in contrast to these well-established methods. By directly comparing the proposed correlation coefficients and the MCGDM method with these techniques, the authors would have the opportunity to elucidate specific scenarios where their method may offer superior results or perhaps identify situations where it may not be the optimal choice. This comparative discussion would significantly enhance the manuscript's academic rigor and provide readers with a clearer understanding of the proposed method's place within the broader context of MCDA research. Furthermore, the research would benefit substantially from detailing the process of constructing the decision matrix and justifying the selection of the criteria involved. There's also an apparent need for the research to augment its contribution by, for instance, employing sensitivity analysis to assess the robustness of the proposed MCGDM method. Additionally, expanding the number of alternatives considered in the case study would provide a more thorough examination of the method's applicability and reliability. Finally, the introduction section demands elaboration to set the stage for the readers properly, providing them with a firm grasp of the research context, the prevailing challenges, and the envisioned solutions.

I suggest a major revision.

Reviewer #2: The paper presents a concept based on correlation coefficients under probabilistic hesitant fuzzy environment and its application in clinical comprehensive evaluation of orphan drugs

Although the concept is potentially interesting, it is unfortunately not translated into a strong methodological and practical contribution.

- The authors do not provide sufficient motivation for the study

- The authors do not benchmark their approach against reference approaches

Once these shortcomings have been remedied, the paper can be re-evaluated

Reviewer #3: This paper proposes a new MCGDM method through improved correlation coefficients under probabilistic hesitant fuzzy environment to evaluate orphan drugs. In my opinion, this work has some merits. I have some suggestions:

1. The literature review part is not just list literatures, you should find the research gap and the implications of your research through the literature review part. However, I can’t see it. I suggest author also should clarify the limitations of existing literatures more clearly, list as 1,2,3….Besides, I suggest adding a separate literature review section.

2. Although this article has been a comprehensive overview, Some classic methods should be mentioned, such as Best-Worst method (BWM), Weighted Aggregated Sum-Product Assessment (WASPAS), SMART (Simple Multi-attribute Rating Technique), DEMATEL (Decision-Making Trial and Evaluation Laboratory),etc. I suggest that the author needs to add relevant content to discuss the reasons why chose to use correlation coefficients this method.

3. The comparison analysis between the proposed method and the existing method and the discussion of the results should be more in-depth.

4. I noticed that some of the references were not convincing enough and suggested updating them.

5. In the conclusion part, the limitations of this paper need to be discussed

6. PLOS authors have the option to publish the peer review history of their article (what does this mean?). If published, this will include your full peer review and any attached files.

Reviewer #1: No

Reviewer #2: No

Reviewer #3: No

---

## [Author Response · Author response to Decision Letter 0]

17 Jan 2024

Dear Reviewers:

Reviewer#1, Concern # 1: The language requires meticulous proofreading by a native speaker to meet the academic standards expected for publication.

Author response: We gratefully appreciate for your valuable suggestion.

Author action: According to the reviewer’s comment, We invited native English speaking editors to help us correct spelling and grammar mistakes in the article. Thank you again for your valuable comment on the paper.

Reviewer#1, Concern # 2: Moreover, the article lacks a distinctive presentation of novelty; there is a noted absence of a compelling argument that clearly defines what sets this research apart from existing studies, a delineation of its unique contributions, and a discussion of its limitations.

Author response: We gratefully appreciate for your valuable suggestion.

Author action: According to the reviewer’s comment, We divided the introduction and the literature review into two parts. In the introduction, we added a summary of the research gaps in existing studies. Then, in the literature review, we reviewed and discussed each research gap separately, aiming at the different research gaps summarized in the introduction. In addition, the existing research gaps are discussed, specifically in the first paragraph of the second part of the paper. Then, in terms of the research on the correlation coefficient of the existing probabilistic hesitancy fuzzy sets, we have discussed the shortcomings of the existing research in the paper before. Finally, in terms of the research on the multi-attribute decision making method, we added the literature review on the MCDM method. The existing MCDM methods are summarized and classified, and then the reasons for selecting the multi-attribute group decision making method based on correlation coefficient are discussed.

Reviewer#1, Concern # 3: The manuscript should offer a more expansive comparison to contemporary methods such as "PT-TOPSIS methods for multi-attribute group decision making under single-valued neutrosophic sets" or "EDAS method for multiple attribute group decision making under spherical fuzzy environment," highlighting both similarities and differences. In addition, a broader literature review and a more comprehensive background on MCDA would fortify the study's contextual relevance and scholarly depthAuthor response: We gratefully appreciate for your valuable suggestion.

Author action: According to the reviewer’s comment, We reviewed and classified the two literatures in the fourth paragraph of the second part

Reviewer#1, Concern # 4: The inclusion of a succinct comparison with methodologies such as SPOTIS, ESP-COMET, SIMUS, RANCOM, and others, is essential to demonstrate the robustness and relevance of the proposed MCGDM technique.

Author response: We gratefully appreciate for your valuable suggestion.

Author action: According to the reviewer’s comment, The main common advantage of the above mentioned methods is that they can overcome the phenomenon of rank reversal. In order to show that the ordering inversion phenomenon does not exist in our method, we add a test on the phenomenon of rank reversal in the seventh part to demonstrate the reliability of our proposed method.

Reviewer#1, Concern # 5: Furthermore, the research would benefit substantially from detailing the process of constructing the decision matrix and justifying the selection of the criteria involved.

Author response: We gratefully appreciate for your valuable suggestion.

Author action: According to the reviewer’s comment, We have added a description of attributes in Table 3 of Part VI. Second, the method decision process has been described in detail in previous articles. 

Reviewer#1, Concern # 6: employing sensitivity analysis to assess the robustness of the proposed MCGDM method.

Author response: We gratefully appreciate for your valuable suggestion.

Author action: According to the reviewer’s comment, We added sensitivity analysis on attribute weights in 7.1 of Part 7 of the paper

Reviewer#1, Concern # 7: Additionally, expanding the number of alternatives considered in the case study would provide a more thorough examination of the method's applicability and reliability.

Author response: We gratefully appreciate for your valuable suggestion.

Author action: According to the reviewer’s comment, In the seventh part of this paper, 7.2, we test whether there is the phenomenon of rank reversal in our method by adding more alternatives, so as to show that our method will not affect the order of the previous schemes when adding alternatives, and thus demonstrate the reliability of our method.

Reviewer#1, Concern # 8: Finally, the introduction section demands elaboration to set the stage for the readers properly, providing them with a firm grasp of the research context, the prevailing challenges, and the envisioned solutions.

Author response: We gratefully appreciate for your valuable suggestion.

Author action: According to the reviewer’s comment, We divided the introduction and the literature review into two parts, in which we added a summary of the research gaps existing in the existing research and provided a brief summary of the work we had done.

Reviewer#2, Concern # 1: The authors do not provide sufficient motivation for the study

Author response: We gratefully appreciate for your valuable suggestion.

Author action: According to the reviewer’s comment, We divided the introduction and the literature review into two parts. In the introduction, we added a summary of the research gaps existing in the existing studies. Then in the literature review, we reviewed and discussed each research gap according to the different research gaps summarized in the introduction

Reviewer#2, Concern # 2: The authors do not benchmark their approach against reference approaches

Author response: We gratefully appreciate for your valuable suggestion.

Author action: According to the reviewer’s comment, in the seventh part, we added sensitivity analysis to our method and tested the ordering inversion phenomenon by referring to other decision-making methods that can effectively overcome the ordering inversion phenomenon. At last, we compared our method with the existing correlation coefficient and the MCDM method based on the correlation coefficient.

Reviewer#3, Concern # 1: 1. The literature review part is not just list literatures, you should find the research gap and the implications of your research through the literature review part. However, I can’t see it. I suggest author also should clarify the limitations of existing literatures more clearly, list as 1,2,3….Besides, I suggest adding a separate literature review section.

Author response: We gratefully appreciate for your valuable suggestion.

Author action: According to the reviewer’s comment, We divided the introduction and the literature review into two parts. In the introduction, we added a summary of the research gaps in existing studies. Then, in the literature review, we reviewed and discussed each research gap separately, aiming at the different research gaps summarized in the introduction. In addition, the existing research gaps are discussed, specifically in the first paragraph of the second part of the paper. Then, in terms of the research on the correlation coefficient of the existing probabilistic hesitancy fuzzy sets, we have discussed the shortcomings of the existing research in the paper before. Finally, in terms of the research on the multi-attribute decision making method, we added the literature review on the MCDM method. The existing MCDM methods are summarized and classified, and then the reasons for selecting the multi-attribute group decision making method based on correlation coefficient are discussed.

Reviewer#3, Concern # 2: 2. Although this article has been a comprehensive overview, Some classic methods should be mentioned, such as Best-Worst method (BWM), Weighted Aggregated Sum-Product Assessment (WASPAS), SMART (Simple Multi-attribute Rating Technique), DEMATEL (Decision-Making Trial and Evaluation Laboratory),etc. I suggest that the author needs to add relevant content to discuss the reasons why chose to use correlation coefficients this method.

Author response: We gratefully appreciate for your valuable suggestion.

Author action: According to the reviewer’s comment, We add a literature review of relevant MCDM methods, summarize and classify existing MCDM methods, and then discuss the reasons for our proposed multi-attribute group decision making method based on correlation coefficient

Reviewer#3, Concern # 3: 3. The comparison analysis between the proposed method and the existing method and the discussion of the results should be more in-depth.

Author response: We gratefully appreciate for your valuable suggestion.

Author action: According to the reviewer’s comment, In the seventh part, we added sensitivity analysis to our method and tested the ordering inversion phenomenon by referring to other decision-making methods that can effectively overcome the ordering inversion phenomenon. At last, we compared our method with the existing correlation coefficient and the MCDM method based on the correlation coefficient.

Reviewer#3, Concern # 3: 4. I noticed that some of the references were not convincing enough and suggested updating them.

Author response: We gratefully appreciate for your valuable suggestion.

Author action: According to the reviewer’s comment, In the second part, we have added some new literature to strengthen the argument of our article and updated some of the literature.

Reviewer#3, Concern # 3: 5. In the conclusion part, the limitations of this paper need to be discussed

Author response: We gratefully appreciate for your valuable suggestion.

Author action: According to the reviewer’s comment, In the conclusion, we add the limitations of our method and look forward to the future to address such limitations.

Thank anonymous reviewers for their valuable suggestions.

---

## [Decision Letter · Decision Letter 1]

8 Apr 2024

A novel MCGDM technique based on correlation coefficients under probabilistic hesitant fuzzy environment and its application in clinical comprehensive evaluation of orphan drugs

PONE-D-23-33158R1

Dear Dr. Hu,

We’re pleased to inform you that your manuscript has been judged scientifically suitable for publication and will be formally accepted for publication once it meets all outstanding technical requirements.

Kind regards,

Fausto Cavallaro, PhD

Academic Editor

PLOS ONE

Additional Editor Comments (optional):

The authors addressed the reviewers comments.

Reviewers' comments:

Reviewer's Responses to Questions

**Comments to the Author**

1. If the authors have adequately addressed your comments raised in a previous round of review and you feel that this manuscript is now acceptable for publication, you may indicate that here to bypass the “Comments to the Author” section, enter your conflict of interest statement in the “Confidential to Editor” section, and submit your "Accept" recommendation.

Reviewer #1: All comments have been addressed

Reviewer #2: All comments have been addressed

2. Is the manuscript technically sound, and do the data support the conclusions?

Reviewer #1: Yes

Reviewer #2: Yes

3. Has the statistical analysis been performed appropriately and rigorously? 

Reviewer #1: N/A

Reviewer #2: N/A

4. Have the authors made all data underlying the findings in their manuscript fully available?

Reviewer #1: Yes

Reviewer #2: Yes

5. Is the manuscript presented in an intelligible fashion and written in standard English?

Reviewer #1: Yes

Reviewer #2: Yes

6. Review Comments to the Author

Reviewer #1: The paper has been improved. There are some small edits mistakes which can be removed in the proofreading stage. Therefore, it can be accepted in its current form.

Reviewer #2: The Authors made great effort to improve the manuscript

I found my previous suggestions adressed

I suggest to accept the paper

7. PLOS authors have the option to publish the peer review history of their article (what does this mean?). If published, this will include your full peer review and any attached files.

Reviewer #1: No

Reviewer #2: No
